# Nkx2.5 marks angioblasts that contribute to hemogenic endothelium of the endocardium and dorsal aorta

Lyad Zamir[1†], Reena Singh[2†], Elisha Nathan[1], Ralph Patrick[2], Oren Yifa[1], Yfat Yahalom-Ronen[1], Alaa A Arraf[3], Thomas M Schultheiss[3], Shengbao Suo[4], Jing-Dong Jackie Han[4], Guangdun Peng[5], Naihe Jing[5], Yuliang Wang[6], Nathan Palpant[7], Patrick PL Tam[8,9], Richard P Harvey[2,10*], Eldad Tzahor[1*]

[1]Department of Molecular Cell Biology, Weizmann Institute of Science, Rehovot, Israel; [2]Victor Chang Cardiac Research Institute, Sydney, Australia; [3]Department of Genetics and Developmental Biology, Rappaport Faculty of Medicine, Technion-Israel Institute of Technology, Haifa, Israel; [4]Key Laboratory of Computational Biology, Chinese Academy of Sciences-Max Planck Partner Institute for Computational Biology, Shanghai Institutes for Biological Sciences, Chinese Academy of Sciences, Shanghai, China; [5]State Key Laboratory of Cell Biology, Institute of Biochemistry and Cell Biology, Shanghai Institutes for Biological Sciences, Chinese Academy of Sciences, Shanghai, China; [6]Institute for Stem Cell and Regenerative Medicine, The University of Washington, Seattle, United States; [7]Institute for Molecular Bioscience, The University of Queensland, Brisbane, Australia; [8]School of Medical Sciences, Sydney Medical School, The University of Sydney, Westmead, Australia; [9]Embryology Unit, Children's Medical Research Institute, Westmead, Australia; [10]St. Vincent's Clinical School, School of Biological and Biomolecular Sciences, University of New South Wales, Kensington, Australia

*For correspondence: r.harvey@victorchang.edu.au (RPH); eldad.tzahor@weizmann.ac.il (ET)

†These authors contributed equally to this work

Competing interests: The authors declare that no competing interests exist.

**Abstract** Novel regenerative therapies may stem from deeper understanding of the mechanisms governing cardiovascular lineage diversification. Using enhancer mapping and live imaging in avian embryos, and genetic lineage tracing in mice, we investigated the spatio-temporal dynamics of cardiovascular progenitor populations. We show that expression of the cardiac transcription factor *Nkx2.5* marks a mesodermal population outside of the cardiac crescent in the extraembryonic and lateral plate mesoderm, with characteristics of hemogenic angioblasts. Extra-cardiac *Nkx2.5* lineage progenitors migrate into the embryo and contribute to clusters of CD41+/CD45+ and RUNX1+ cells in the endocardium, the aorta-gonad-mesonephros region of the dorsal aorta and liver. We also demonstrated that ectopic expression of *Nkx2.5* in chick embryos activates the hemoangiogenic gene expression program. Taken together, we identified a hemogenic angioblast cell lineage characterized by transient *Nkx2.5* expression that contributes to hemogenic endothelium and endocardium, suggesting a novel role for *Nkx2.5* in hemoangiogenic lineage specification and diversification.

## Introduction

Development of the cardiovascular system takes place during the early stages of embryogenesis. Cardiac progenitors residing in the cardiac crescent are formed from the first heart field (FHF) located in the anterior lateral plate mesoderm (LPM). As the embryo develops, FHF progenitors fuse

**eLife digest** As an animal embryo develops, it establishes a circulatory system that includes the heart, vessels and blood. Vessels and blood initially form in the yolk sac, a membrane that surrounds the embryo. These yolk sac vessels act as a rudimentary circulatory system, connecting to the heart and blood vessels within the embryo itself. In older embryos, cells in the inner layer of the largest blood vessel (known as the dorsal aorta) generate blood stem cells that give rise to the different types of blood cells.

A gene called *Nkx2.5* encodes a protein that controls the activity of a number of complex genetic programs and has been long studied as a key player in the development of the heart. *Nkx2.5* is essential for forming normal heart muscle cells and for shaping the primitive heart and its surrounding vessels into a working organ. Interfering with the normal activity of the *Nkx2.5* gene results in severe defects in blood vessels and the heart. However, many details are missing on the role played by *Nkx2.5* in specifying the different cellular components of the circulatory system and heart.

Zamir et al. genetically engineered chick and mouse embryos to produce fluorescent markers that could be used to trace the cells that become part of blood vessels and heart. The experiments found that some of the cells that form the blood and vessels in the yolk sac originate from within the membranes surrounding the embryo, outside of the areas previously reported to give rise to the heart. The *Nkx2.5* gene is active in these cells for only a short period of time as they migrate toward the heart and dorsal aorta, where they give rise to blood stem cells

These findings suggest that *Nkx2.5* plays an important role in triggering developmental processes that eventually give rise to blood vessels and blood cells. The next step following on from this work will be to find out what genes the protein encoded by Nkx2.5 regulates to drive these processes. Mapping the genes that control the early origins of blood and blood-forming vessels will help biologists understand this complex and vital tissue system, and develop new treatments for patients with conditions that affect their circulatory system. In the future, this knowledge may also help to engineer synthetic blood and blood products for use in trauma and genetic diseases.

at the midline to form the primitive heart tube, which begins to beat and, as a consequence, blood begins to circulate (*DeRuiter et al., 1992*; *Stalsberg and DeHaan, 1969*). Second heart field (SHF) progenitors residing within the pharyngeal mesoderm (*Diogo et al., 2015*) contribute to subsequent growth and elongation of the heart tube (*Kelly et al., 2001*; *Mjaatvedt et al., 2001*; *Waldo et al., 2001*). In both chick and mouse embryos, the FHF gives rise to myocytes of the left ventricle and parts of the atria, whereas the SHF contributes to myocardium of the outflow tract, right ventricle, and atria (*Buckingham et al., 2005*). Recent studies suggest that these heart fields contain both uni-potent and multipotent mesodermal progenitors that give rise to the diverse lineage types within the heart (*Kattman et al., 2006*; *Lescroart et al., 2014*; *Meilhac et al., 2004*; *Moretti et al., 2006*; *Wu et al., 2006*). For example, bipotent SHF progenitors generate endocardium or smooth muscle cells as well as cardiomyocytes (*Lescroart et al., 2014*; *Moretti et al., 2006*). Cardiovascular pro-genitors sequentially express the cardiac transcription factors *Mesp1/2, Islet1* (*Isl1*) and *Nkx2.5*, and, in response to cues from the microenvironment, undergo lineage diversification and differentiation (*Laugwitz et al., 2008*; *Prall et al., 2007*; *Saga et al., 1999*).

The formation of blood vessels begins with the appearance of blood islands in the extraembryonic region. In the chick embryo, this occurs in the *area vasculosa* around St. 3–5. Sabin first proposed that some blood cells differentiate directly from endothelial cells (*Sabin, 1920*). Indeed, endothelial and blood cells that form the rudimentary circulatory system have long been thought to originate from bipotent mesoderm progenitors termed 'hemangioblasts' (*Choi et al., 1998*). Recent experimental advances revealed the existence of a specialized cell, hemogenic endothelium, that harbours the potential to generate hematopoietic progenitors (*Boisset et al., 2010*; *Jaffredo et al., 1998*). These cells arise early in embryonic development and migrate from extra- to intra-embryonic locations (*Tanaka et al., 2014*). However, while blood cell formation from hemogenic endothelium has been visualised directly in multiple animal models (*Bertrand et al., 2010*; *Boisset et al., 2010*;

*Kissa and Herbomel, 2010*; *Lam et al., 2010*), and *in vitro* during ES cells differentiation (*Eilken et al., 2009*; *Lancrin et al., 2009*), the existence in vivo of a bipotential hemangioblast that contributes to both blood and endothelial cells remains controversial (*Hirschi, 2012*). Recent lineage tracing and live imaging studies addressing the ontogenic origins of hemoangiogenic cells in mouse embryos suggest that all hemogenic cells derive from a Flk1$^+$/Runx1$^+$/Gata1$^-$ 'hemogenic angioblast' population located in extraembryonic mesoderm bordering the region forming blood islands (*Tanaka et al., 2014*). As for angioblasts that contribute to the embryonic vasculature, these cells actively migrate to embryonic sites of hemopoiesis before the establishment of circulation. Runx1 activation in the aorta-gonad-mesonephros (AGM) region is essential at this early stage for the specification of hemogenic endothelial cell fate and for definitive hemopoiesis (*Tanaka et al., 2012*).

The origins of the myocardium have been studied intensively, although that of the endocardium remains largely obscure (*Harris and Black, 2010*). Early lineage tracing studies in chick have shown that myocardial and endocardial progenitors segregate at primitive streak stages (*Wei and Mikawa, 2000*). In line with this view, we previously demonstrated that the endocardium originates, in part, from committed vascular endothelial progenitors that segregate medially from myocardial progenitors within the cardiac crescent (*Milgrom-Hoffman et al., 2011*). Numerous studies have demonstrated the close ontological relationship between the blood and vascular lineages, and the myocardium and endocardium of the forming heart. Heart progenitor cells in the embryo, or those derived from differentiating embryonic stem cells, co-express markers of blood (*Tal1*, *Gata1*) and endothelial (*Flk1*) lineages; conversely, vascular progenitors in the yolk sac express at some point in their lineage histories multiple cardiac transcription factors including Nkx2.5, Isl1 and Tbx20 (*Keenan et al., 2012*; *Stanley et al., 2002*; *Stennard et al., 2005*). Studies in both fish and mouse show that the blood and heart-forming territories develop in close proximity, and are mutually antagonistic (*Bussmann et al., 2007*; *Schoenebeck et al., 2007*; *Van Handel et al., 2012*). These territories appear to represent distinct spatio-temporal domains of the nascent mesoderm regulated in common by the transcription factor Mesp1 (*Chan et al., 2013*).

*Nkx2.5* has long been known as a key transcription factor for cardiac development, specifically in the specification and differentiation of the myocardial lineage (*Evans et al., 2010*; *Lyons et al., 1995*; *Turbay et al., 1996*). However, the mis-patterning of the heart and arrest of cardiac development in *Nkx2-5* null mutant mouse embryos has precluded a deep analysis of the role of this transcription factor in other mesodermal lineages. Cre recombinase-based lineage tracing with *Nkx2-5$^{irescre}$* driver mice revealed a contribution of labelled cells to endothelial and blood cells in the yolk sac blood islands (*Stanley et al., 2002*), consistent with severe defects reported in remodelling of the yolk sac vasculature in *Nkx2-5* null embryos (*Tanaka et al., 1999*). However, it was not determined whether this was a primary affect of loss of Nkx2-5 in yolk sac, or secondary to hemodynamic effects arising from arrested heart development.

Recent studies have begun to shed new light on the role of *Nkx2-5* in hemoangiogenic lineage specification. Nkx2-5 was found to directly bind a cis-regulatory element of the *Etv2* gene, encoding an ETS domain family transcription factor essential for formation of the embryonic and extra-embryonic endothelium, endocardium and blood lineages, and Nkx2-5 induces its expression in vitro (*Ferdous et al., 2009*). In zebrafish, clusters of *nkx2.5$^+$* cells segregate from the main *nkx2.5$^+$* myogenic fields prior to heart formation, giving rise to the pharyngeal arch arteries (*Paffett-Lugassy et al., 2013*). Here, *nkx2.5* is essential for endothelial cell fate specification and expression of *etsrp/etv2* and *scl/tal1*. Consistent with these results, genetic lineage tracing in the mouse using the *Nkx2-5$^{irescre}$* driver showed a contribution of Nkx2-5$^+$ cells to the pharyngeal arch arteries and aortic sac, and analysis of *Nkx2-5* null embryos demonstrated that Nkx2-5 is required for pharyngeal arch artery formation (*Paffett-Lugassy et al., 2013*). Nkx2-5 and Isl1 are expressed in and required for a distinct subset of endocardial cells defined as the hemogenic endocardium, proposed to contribute to primitive and transient definitive hematopoiesis in the embryo (*Nakano et al., 2013*). Finally, expression of the human *NKX2-5* gene occurs in a subset of paediatric T and B cell acute lymphoblastic leukaemias defined by a t(5;14) translocation (*Nagel et al., 2003*; *Su et al., 2008*).

Despite this knowledge, the origins of hemogenic endothelium and endocardium, and the role of Nkx2.5 in their lineage networks in vertebrates, are poorly understood. In this study, we used chick and mouse embryonic models to investigate the origins and plasticity of early cardiovascular progenitors. Using defined early *Nkx2-5* and *Isl1* embryonic enhancers driving expression of GFP and RFP as surrogate lineage tracing tools in the chick, we revealed the origin of the cardiac hemogenic

endocardium to be hemogenic angioblasts in the extraembryonic/lateral plate mesoderm. These mesodermal cells express *Nkx2.5* already in the primitive streak before migrating to the posterior LPM and extraembryonic regions, and, subsequently, into the endocardium and DA, where they form hemogenic endothelial clusters. We further demonstrated in chick, through gain-of-function experiments, that Nkx2.5 induces the expression of hemoangiogenic markers in nascent mesoderm. In mouse, *Nkx2-5*[irescre] lineage tracing demonstrated activation of *Nkx2-5* in extraembryonic tissue proximal to the cardiac crescent, and Nkx2-5[+] cells make a substantial contribution to Runx1[+] and CD41[+] hemogenic endothelium in the DA and to a lesser extent endocardium. Our data provide the first in vivo evidence that Nkx2.5 has a role in initiating the hemoangiogenic program and that Nkx2.5[+] progenitors contribute via the LPM/extraembryonic region to hemogenic endothelium of the endocardium and AGM region, and subsequently blood. Furthermore, we identify a novel population of endocardial progenitors outside the classical FHF and SHF domains.

## Results

### Nkx2.5 and Isl1 enhancers serve as lineage reporters highlighting spatiotemporal cardiovascular cell dynamics in live embryos

The chick embryo is highly suitable for the study of early development due to its accessibility for in vivo manipulations such as tissue grafting and cell fate mapping. In order to investigate the spatiotemporal dynamics of early cardiogenesis, we combined bioinformatic enhancer identification with DNA electroporation and time-lapse imaging of enhancer reporters in live chick embryos cultured *ex ovo* (*Uchikawa et al., 2004*). We first analyzed the expression in chick embryos of a previously characterized 513 bp early mouse *Nkx2-5* cardiac enhancer (*Lien et al., 1999*) (*Nkx2*-5-en; *Figure 1A–E*; n = 30). Despite apparent lack of sequence conservation with the chick *Nkx2.5* locus, *Nkx2-5*-en was able to drive robust reporter gene expression in cardiac progenitors in live chick embryos (*Figure 1B–C*). *Nkx2-5*-en expression corresponded well with the mesoderm expression domain of endogenous chick *Nkx2.5* at embryonic St. 8 (*Figure 1E–E'''*). We then identified a conserved 830 bp regulatory element flanking the mouse genomic locus of the *Isl1* gene, which acts as an early cardiac enhancer in chick embryos (*Isl1*-en; *Figure 1F–J'*; n = 30, *Video 1*; n = 3). The expression of *Isl1*-en recapitulated the endogenous mesoderm and endoderm (RNA and protein) expression domains of chick Isl1 (*Figure 1I–J*), but was absent from ectoderm (*Figure 1J'* and *Figure 1—figure supplement 1*).

### Cardiac development relies on distinct cardiovascular populations

To study the dynamics of cardiac progenitor specification and migration in vivo, we co-electroporated *Isl1* and *Nkx2-5* enhancers into primitive streak-stage chick embryos (*Figure 2A*). Embryos were incubated overnight and monitored from St. 7 onwards. Live-cell imaging revealed the presence of both common and distinct cell populations within the cardiac crescent and adjacent developing *area vasculosa* (*Figure 2B*; n = 30, *Video 2*; n = 3). At St. 7–8, *Nkx2-5*-en-GFP and *Isl1*-en-RFP double-positive cells (yellow) migrated to the anterior and lateral regions of the embryo and contributed to the cardiac crescent (*Figure 2B,C*). However, a small proportion of electroporated cells expressed either GFP or RFP alone (*Figure 2B–E*). The anterior part of the heart tube (outflow region) preferentially contained GFP[+] (Nkx2.5[+]) cardiac progenitors, while the posterior part (inflow) was mostly RFP[+] (Isl1[+]) (*Figure 2B–E*). This is consistent with studies showing that the myocardium of the inflow pole of the mouse heart can form from Isl1[+] progenitors in the absence of Nkx2.5 expression (*Christoffels et al., 2006*).

By St. 9–10, FHF/cardiac crescent cells have already fused into the primitive heart tube, which continues to grow with the addition of SHF cells at its inflow and outflow poles (*Figure 2C–E*). In the chick and mouse, Nkx2.5 is expressed throughout cardiac development, while Isl1 expression is transient and progressively downregulated as heart progenitors differentiate into specialised cardiomyocytes (*Nathan et al., 2008*). Consistently, the middle part of the primary heart tube exhibited stronger GFP expression (*Figure 2E*). Surprisingly, we could detect a trail of GFP[+]/RFP[-] (*Nkx2-5*-en[+]-*Isl1*-en[-]) cells extending from the LPM and extraembryonic regions towards the heart (*Figure 2D*, green arrowheads). This expression was not observed for *Isl1*-en-RFP, which was restricted to the primitive heart tube including both outflow and inflow tracts. Further time-lapse live imaging

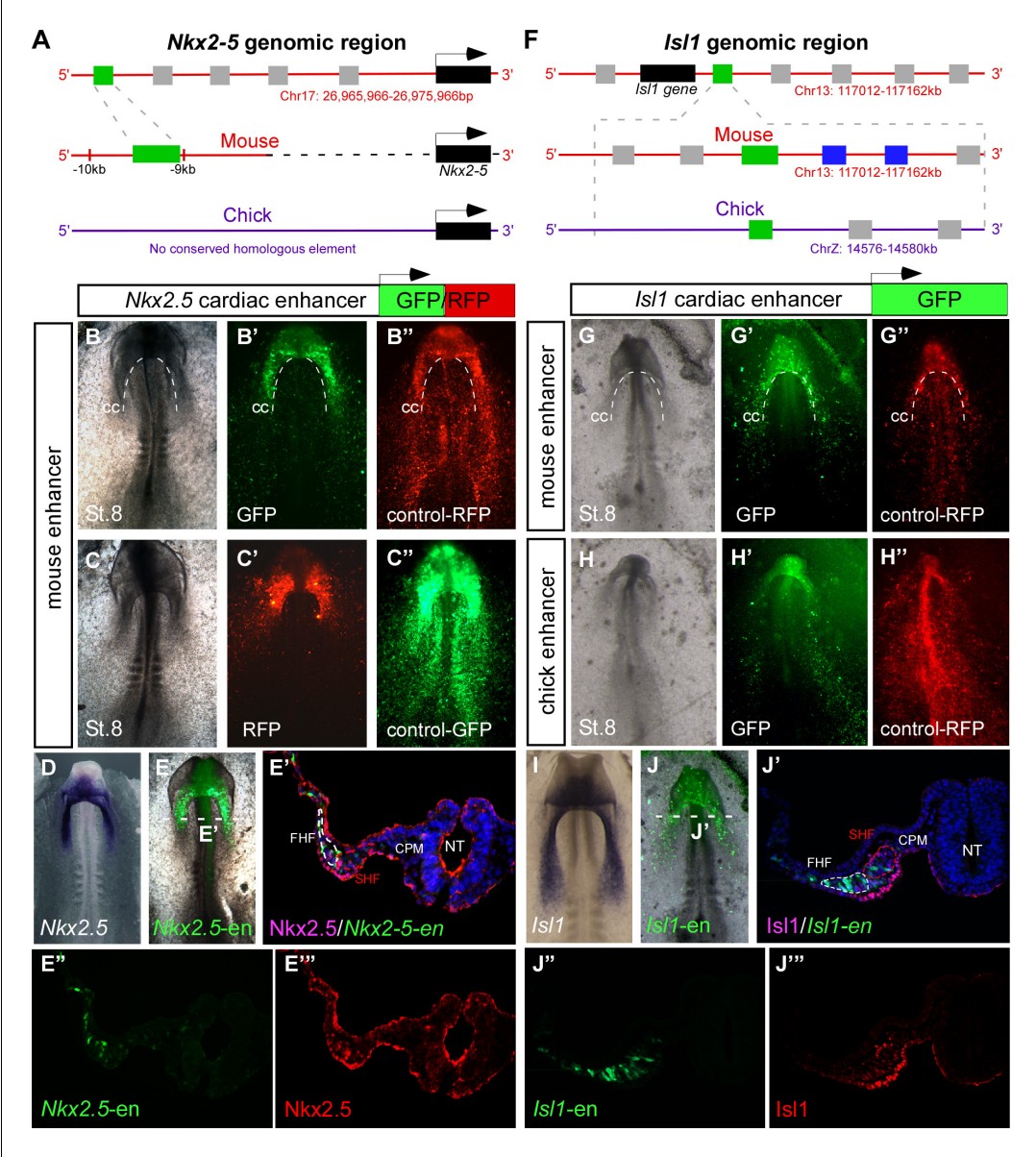

**Figure 1.** Identification of novel *Nkx2-5* and *Isl1* cardiac enhancers. (**A**) Location of the *Nkx2-5* enhancer in the mouse genome. (**B–C**) Expression patterns of the enhancer-driven GFP/RFP formed in the cardiac crescent at St. 8, compared to corresponding control vectors. (**D**) *In situ* hybridization of endogenous *Nkx2.5* compared to (**E**) the mouse *Nkx2.5* enhancer. (**E'–E'''**) Cross-section of the electroporated embryo after immunostaining for Nkx2.5 (magenta), GFP (green), and DAPI (blue). Separated channels for the *Nkx2-5*-en (**E"**) and Nkx2.5 protein (**E'''**). (**F**). Location of the *Isl1* enhancer in chick and mouse genomes. (**G–H**) Expression patterns of the enhancer-driven GFP formed in the cardiac crescent at St. eight for both chick and mouse elements. The control RFP vector is expressed in all cells. (**I**) *In situ* hybridization of endogenous *Isl1*, compared to (**J**) stage-matched mouse enhancer-electroporated embryo. (**J'–J'''**) Cross-section of the electroporated embryo after immunostaining for Isl1 (magenta), GFP (green), and DAPI (blue). Separated channels for the mouse *Isl1*-en (**J"**) and the Isl1 protein (**J'''**). **B-E**, n = 30; **F-J**, n = 30. FHF: first heart field; SHF: second heart field; CPM: cranial paraxial mesoderm; NT: neural tube; CC: cardiac crescent. See also *Figure 1—figure supplement 1*, *Video 1*.

The following figure supplement is available for figure 1:

**Figure supplement 1.** Characterization of the novel cardiac *Isl1* enhancer.

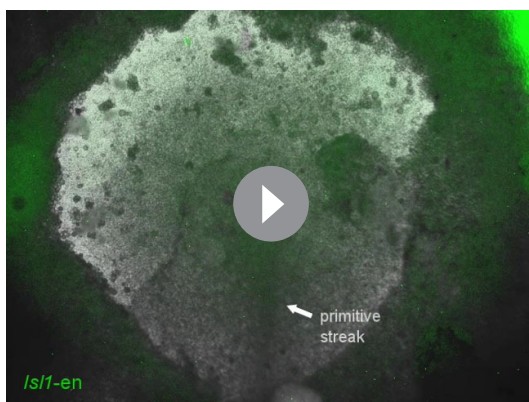

**Video 1.** Time lapse movie of a developing embryo expressing the *Isl1* enhancer. Relates to *Figure 1*. An embryo electroporated with the *Isl1*-en (GFP) plasmid at St. 3. The time-lapse video follows from St.5 to St.11 when the heart tube has already formed and the embryo has begun to turn. oft: outflow; inf: inflow. DOI: 10.7554/eLife.20994.005

suggested that *Nkx2-5*-en⁺ (*Isl1*-en⁻) progenitors migrate into the forming heart via the inflow tract (*Figure 2D*, *Figure 2—figure supplement 1*, *Video 2*). Taken together, our findings reveal the existence of cardiac progenitors within and outside the cardiac crescent that differ in their *Isl1* and *Nkx2.5* spatiotemporal expression patterns.

## The Nkx2-5 enhancer marks hemogenic angioblast progenitors

Our dynamic enhancer imaging suggests that *Nkx2-5*-en-GFP marks a broad mesoderm population with both cardiac and angioblastic characteristics. The GFP⁺ angioblast progenitors form a continuum between the cardiac crescent defined by *Isl1*-en-RFP, and LPM and extraembryonic mesoderm, and are incorporated into the forming heart tube. To explore this idea further, we used a hemangioblast enhancer (Hb-en) located in the cis-regulatory region of the chick *Cerberus* gene, which marks blood islands and migratory angioblasts in the yolk sac (*Teixeira et al., 2011*). Co-electroporation of *Nkx2-5*-en-RFP and Hb-en-GFP enhancers at St. 3 with analysis at St. 7 revealed *Nkx2-5-en*-RFP⁺ progenitors within the cardiac crescent in a pattern similar to that of *Nkx2.5* mRNA, as well as in LPM and extraembryonic mesoderm (*Figure 3A–B*). Hb-en-GFP was expressed only in posterior LPM/extraembryonic mesoderm bordering the *area vasculosa* of the yolk sac (*Figure 3A–C*; n = 17/20). Many cells co-expressed *Nkx2-5*-en-RFP and Hb-en-GFP within the LPM (*Figure 3A'''*, yellow arrowheads). The *Nkx2-5*-en expression domain in the LPM/extraembryonic region overlapped with that of the endothelial/blood marker *Tal1* (*Figure 3C*).

We next explored the ability of Hb-en⁺ and *Nkx2-5*-en cells to generate vascular endothelial and hemogenic cells in vitro by culturing posterior LPM explants following the electroporation of the two reporters (*Figure 3—figure supplement 1*). GFP⁺ and RFP⁺ cells in LPM explants expressed endothelial (Cdh5) and hemogenic (Runx1 and CD45) mesodermal markers (*Figure 3—figure supplement 1*), suggesting that Hb-en⁺/*Nkx2-5*-en⁺ double positive enhancer labels hemogenic angioblasts located in the yolk sac and LPM. Based on the anatomical location, we hypothesize that *Nkx2-5*-en/Hb-en double positive cells represent hemogenic angioblasts (*Tanaka et al., 2014*; *Teixeira et al., 2011*) and that *Nkx2-5*-en marks distinct cardiomyocyte and hemogenic angioblast populations in the chick embryo.

## Hemogenic angioblasts migrate into the heart through the inflow tract and contribute to the endocardium

We next assessed the contribution of Hb-en⁺ hemogenic angioblasts to the developing heart tube by electroporating Hb-en-GFP at St. 3, with analysis at St.6–11 (*Figure 4A–G*). Live-imaging analysis revealed GFP⁺ cells outside the cardiac crescent at St.6 that were eventually incorporated into the heart at St.11 (*Video 3*; n = 3). These cells or their daughter cells (*Figure 4A–G*, orange and blue arrowheads) migrated towards and incorporated into the venous pole of the heart (*Figure 4B–E*), while some daughters remained in the vitelline vasculature.

We next co-electroporated embryos with both *Nkx2-5*-en and Hb-en plasmids and followed the fate of double-labelled cells. Migration of Hb-en⁺/*Nkx2-5*-en⁺ cells towards the heart and yolk sac could be tracked (*Figure 4—figure supplement 1*, *Video 4*). Histological sections showed that expression of Hb-en (GFP⁺, green arrow) was restricted to the endocardium (*Figure 4H–I*), while *Nkx2-5*-en expression (RFP⁺, red arrow) was detected in both myocardium and endocardium (*Figure 4H*; n = 12). Importantly, *Nkx2-5*-en⁺/Hb-en⁺ (yellow) cells were found only in the endocardium (*Figure 4H*, yellow arrows; *Figure 4—source data 1*). Nkx2.5 protein was expressed only in

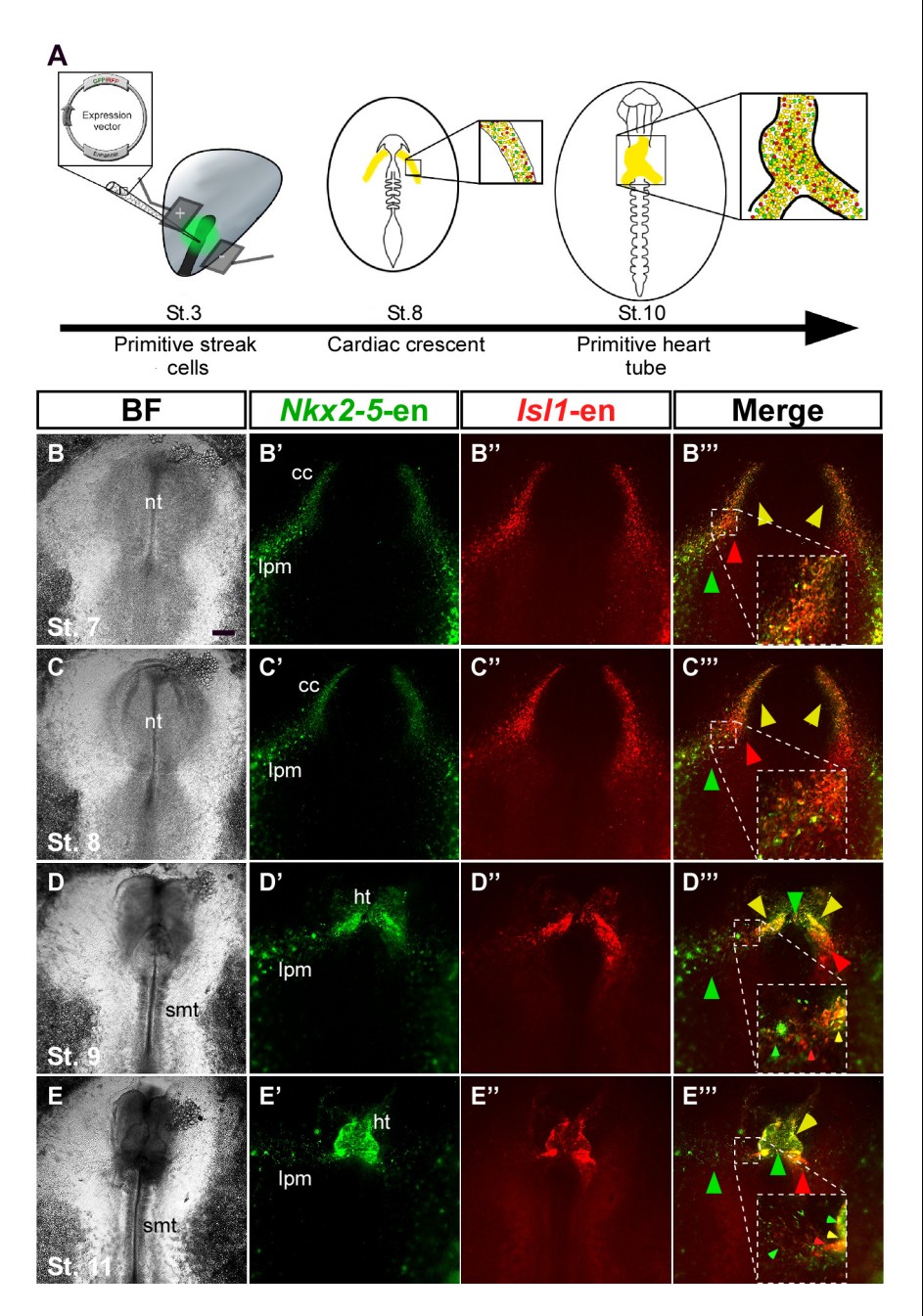

**Figure 2.** *Nkx2-5* and *Isl1* enhancers mark distinct cardiac progenitor populations. (**A**) Experimental design of the *ex ovo* electroporation technique. An electric field is used to introduce different plasmids into the embryo at gastrulation (St. 3), followed by EC culture and imaging. (**B–E**) Images taken from a 24 hr time-lapse video of an embryo co-electroporated with the *Nkx2-5* (GFP) and *Isl1* (RFP) enhancers from St. 7 to St. 11. Higher magnification panels (**B'''–E'''**) of the dotted square boxes highlight the differential *Nkx2-5/Isl1* enhancers expression. Green arrow: *Nkx2-5*-en[+] cells; Red arrow: *Isl1*-en[+] cells; Yellow arrow: *Nkx2-5*-en[+]/*Isl1*-en[+] cells. CC: cardiac crescent; LPM: lateral plate mesoderm; HT: heart tube; smt: somites. Scale bar: 100 μm. See also *Video 2*.

The following figure supplement is available for figure 2:

**Figure supplement 1.** Migration of *Nkx2-5*-en[+] cells from outside the cardiac crescent towards the heart tube.

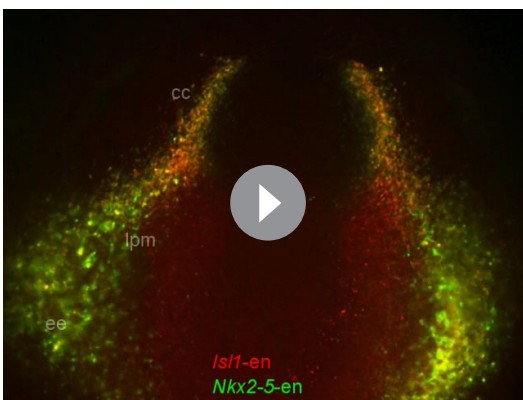

**Video 2.** Time lapse movie of a developing embryo co-expressing the *Nkx2-5* and *Isl1* enhancers. Relates to *Figure 2*. A St. 3 embryo was co-electroporated with the *Isl1*-en (RFP) and *Nkx2-5*-en (GFP) plasmids. The embryo was cultured overnight and time-lapse analysis started at St.6. The embryo was allowed to grow to St.11–12 when the heart tube is visible and begins to loop. The BF channel was omitted from the movie to enhance the visibility of distinct cardio-vascular progenitor populations. cc: cardiac crescent; lpm: lateral plate mesoderm; ee: extraembryonic; da: dorsal aorta; shf: second heart field.

the myocardium at this stage (*Figure 4I*), indicating that *Nkx2-5*-en marks cells expressing Nkx2.5 protein at an earlier progenitor stage.

To quantify the contribution of hemogenic angioblasts to the endocardium, Hb-en-GFP was co-electroporated with a control *pCAGG*-RFP vector as a reference for electroporation efficiency. At St. 12, a large number of cells across the heart were *pCAGG*-RFP[+], while Hb-en-GFP[+] cells could be detected in a salt and pepper pattern (*Figure 4J*). Embryos were subsequently cryo-sectioned at the level of the heart tube and cells positive for both GFP and RFP were quantified separately in the myocardium and endocardium (*Figure 4J'–J"*). As expected, the contribution of Hb-en[+] cells to the myocardium was <0.1% (*Figure 4K*). In the endocardium, ~15% of electroporated pCAGG-RFP[+] cells expressed Hb-en-GFP (*Figure 4J–J"*, *L*). Further analysis revealed that the Hb-en[+] contribution was graded slightly but significantly between the two poles of the heart - from 13% at the arterial pole (outflow tract) to 18% at the venous pole (inflow tract) where the heart connects with the vitelline veins (*Figure 4M*). Taken together, our results reveal a novel population of cardiovascular progenitors located in the LPM/extraembryonic region outside the cardiac crescent, which contributes specifically to the endocardium. This progenitor population is distinct from previously characterized SHF endocardial progenitors located medial to myocardial progenitors in the cardiac crescent (*Milgrom-Hoffman et al., 2011*).

## Hb-en[+] and Nkx2-5-en[+] cells contribute to the hemogenic endothelium of the dorsal aorta

Formation of the DA involves the integration of early endothelial cells from the LPM (*Sato, 2013*). We next determined whether Hb-en[+]/*Nkx2-5-en[+]* cells contribute to the endothelial layer of the DA. To visualize endothelium as well as hemogenic endothelium in the DA, we used our enhancers in conjunction with a set of markers including VE-cadherin (Cdh5), CD45 and Runx1 (*Figure 5*). While Cdh5 labelled the entire DA endothelium, CD45 expression was restricted to hemogenic clusters in the floor of the DA at St. 17 (*Figure 5A*). To investigate the contribution of Hb-en[+] cells to the hemogenic endothelium, embryos were electroporated with Hb-en and cultured for 3 days until St. 14–15. Hb-en expression was detected in the endothelial layer of the floor of the DA (n = 4/6), as well as in the endocardium (n = 6/6) but not in the cardinal vein (*Figure 5B*, *Figure 4—source data 1*). Co-expression of the Hb-en-GFP with CD45 and Runx1 revealed hemogenic clusters in the DA, highlighting the hemogenic angioblast origin of the hemogenic endothelium (*Figure 5B*), in accordance with previous findings (*Lancrin et al., 2009*; *Tanaka et al., 2014*). We propose that hemogenic cells are incorporated into the floor of the DA during its formation from LPM.

We also found *Nkx2-5*-en[+] cells in the endothelial layer of the DA that co-expressed Cdh5 (*Figure 5C*, white arrows; n = 5/7). Earlier, at St.13–14, these cells were localized at the lateral edge of the newly fused DA and failed to express CD45, although, Runx1, an early hematopoietic stem cell marker known to be activated in hemogenic cells of the DA (*North et al., 2002*), was co-expressed with *Nkx2-5*-en[+] cells at this stage (*Figure 5D*). These results suggest that *Nkx2.5*[+] hemogenic angioblasts contribute to the endothelial layer and hemogenic endothelium of the DA during its formation.

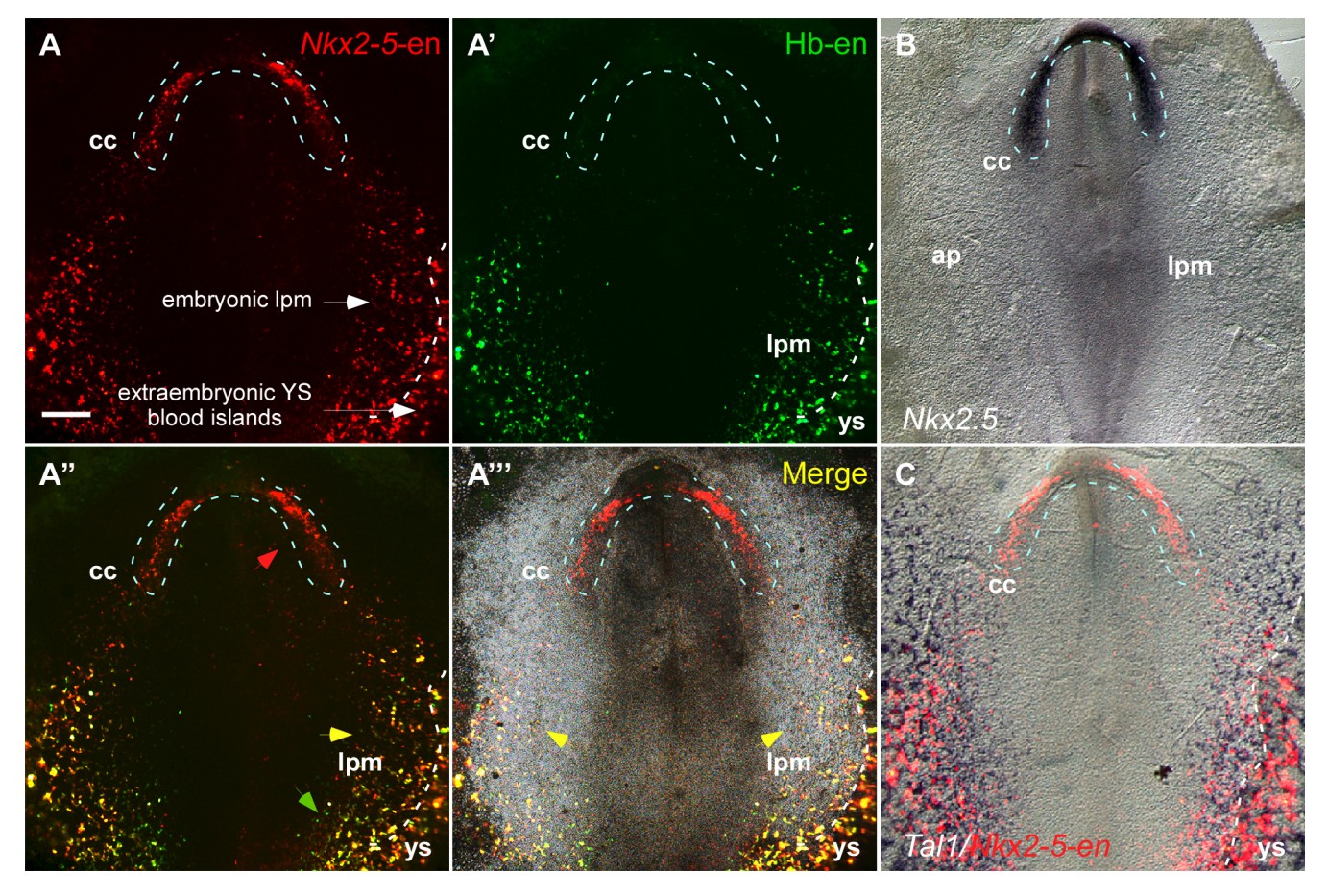

**Figure 3.** The *Nkx2-5* enhancer is expressed in a hemangiogenic cell population. (**A–A'''**) A St. 7 embryo co-electroporated with the *Nkx2-5*-en (RFP) and Hb-en (GFP). The dashed line marks the boundaries of the cardiac crescent. Red arrows represent *Nkx2-5*-en[+] cells in the cardiac crescent; yellow arrows represent *Nkx2-5*-en[+]/Hb-en[+] cells; green arrows represent Hb-en[+] cells (**B**) *In situ* hybridization for the endogenous *Nkx2.5* mRNA of a St. 7 embryo. (**C**) *In situ* hybridization for the endogenous *Tal1* mRNA of a St. 7 embryo. *Tal1* expression pattern is compared to a representative *Nkx2-5*-en expression pattern from different embryo. CC: cardiac crescent; AP: area pellucida; LPM: lateral plate mesoderm; YS: yolk sac. Scale bar: 100 µm. n = 17/20. See also *Figure 3—figure supplement 1*.

The following figure supplement is available for figure 3:

**Figure supplement 1.** The Hb enhancer is active in endothelial and blood progenitors.

## Hb-en[+] and Nkx2-5-en[+] cells give rise to hemogenic endocardium

We next sought to investigate the contribution of Hb-en[+] and *Nkx2-5*-en[+] cells to hemogenic endocardium, recently reported to derive from the *Nkx2-5*[+] lineage in the mouse (*Nakano et al., 2013*). Currently, the origin of these cells is unknown. Endocardial cells co expressing *Nkx2-5*-en[+] and Cdh5[+] could be detected in St.13–14 chick embryos (*Figure 6A*) and we found that a subset of these cells had the same marker profile as in the DA (*Figure 6B*). At the level of the developing ventricle, we observed Hb-en-GFP[+] cells co-expressing either CD45 or Runx1, or both (*Figure 6C–D*). Taken together, our data suggest that the hemogenic endothelium within the primitive heart tube of the chick embryo is derived from hemogenic angioblasts within the LPM/extraembryonic mesoderm.

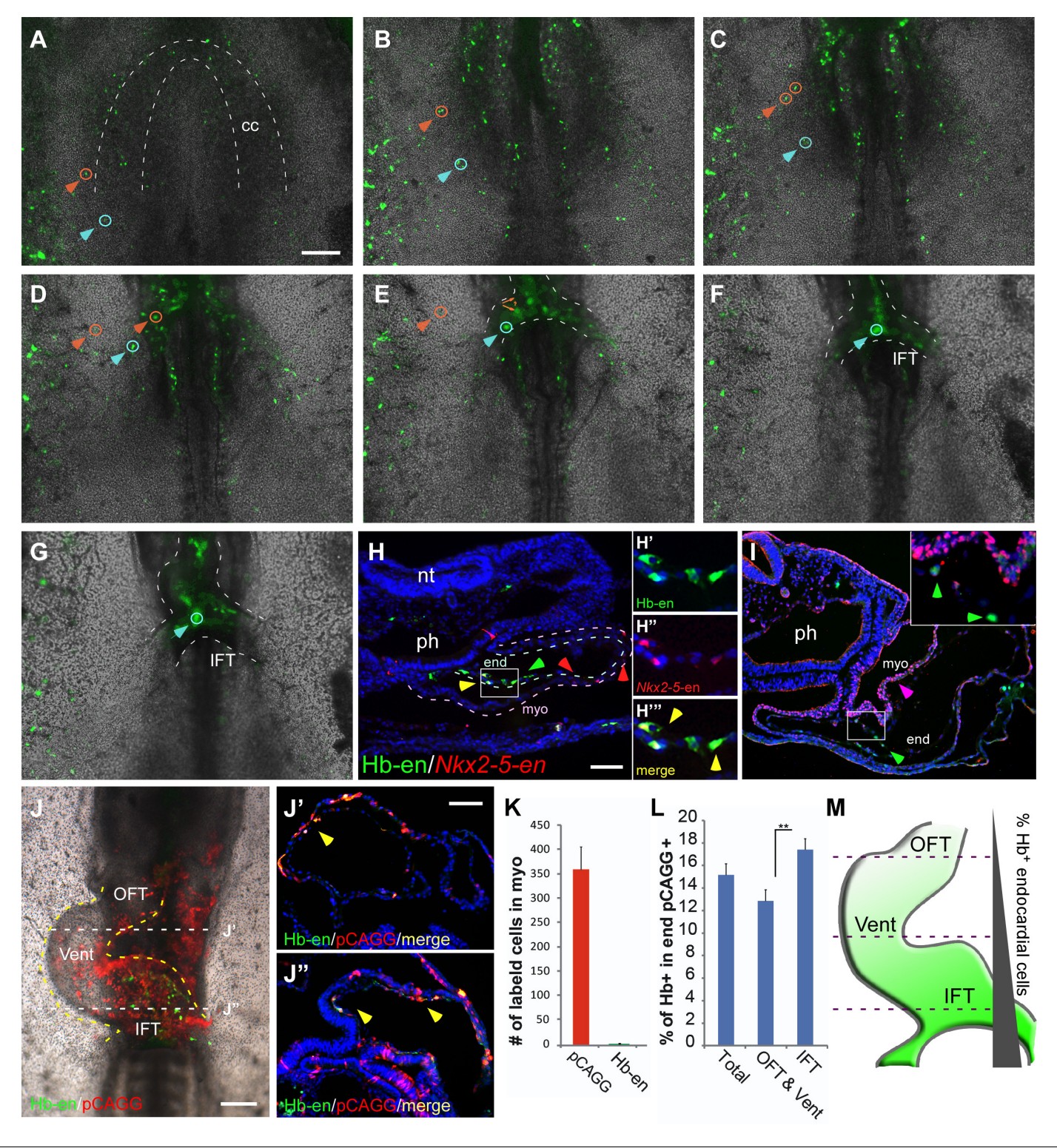

**Figure 4.** Hemangiogenic progenitors migrate to the heart through the inflow tract, and contribute to the endocardium. (A–G) Time-lapse images taken from an embryo electroporated with Hb-en (GFP). Images span a 24 hr time frame, from St. 6 to St. 11 of the same embryo. The orange and cyan circles distinguish between two distinct Hb-en⁺ cells. Cells were manually marked and tracked throughout the 24 hr window. (H) A St. 11 embryo co-electroporated with *Nkx2-5*-en and Hb-en. The embryo was sectioned at the level of the heart to show the distribution of cells within the heart tube. Arrows mark Hb-en⁺ (green), Hb⁻en⁺/*Nkx2-5*-en⁺ (yellow) and *Nkx2-5*-en⁺ (red) cells. (H′–H‴) include higher magnification images of single channels from the square box in H. (I) A St. 11 embryo electroporated with Hb-en. The section was immunostained for Nkx2.5 (magenta), and its expression

*Figure 4 continued on next page*

*Figure 4 continued*

pattern compared to that of Hb-en. (**J**) An embryo electroporated with the Hb-en-GFP and control vector pCAGG-RFP at St. 12. (**J'**) Cryo-section through the outflow tract region (OFT) of the heart shown in J. (**J''**) Cryo-section through the inflow tract region (IFT) of the heart shown in J. Yellow arrows correspond to Hb-en[+]/pCAGG[+] cells in the endocardium. (**K**) Quantification of Hb-en[+] cell contribution to the myocardium compared to the contribution of pCAGG-RFP[+] cells. (**L**) Quantification of Hb-en[+] cell contribution to the endocardium relative to the contribution of pCAGG-RFP[+] cells. The quantifications represent two independent experiments with n = 3. For both quantifications, double positive cells (RFP[+]/GFP[+]) were manually counted and compared to the total number of RFP[+] cells. (**M**) Schematic representation of relative Hb-en[+] cell contribution to different parts of the endocardium. Hb-en[+] cells numbers are increased towards the venous (IFT) pole of the heart. CC: cardiac crescent; IFT: inflow tract; OFT: outflow tract; vent: ventricle; myo: myocardium; end: endocardium. Scale bars: 100 μm. n = 9/12. See also *Video 3* and *Figure 4—source data 1*.

The following source data and figure supplement are available for figure 4:

**Source data 1.** The distribution of Hb-en[+] cells in the chick embryo.

**Figure supplement 1.** Double positive *Nkx2-5*-en[+]/Hb-en[+] cells migrate towards the heart and yolk sac.

## Nkx2.5 is expressed in the nascent mesoderm in the primitive streak of chick embryos

To examine the potential of Nkx2.5 to induce cardiovascular/blood lineage commitment in chick embryos, we first compared the expression profile of key cardiovascular and blood transcription factors at St. 7 (cardiac crescent stage; *Figure 7A*). Regional expression of these factors can be seen along the medial-lateral and anterior-posterior axes in zones generally defined as 'cardiac' (*Tbx5*, *Isl1*, *Nkx2.5*), 'cardiovascular' (*Isl1*, *Nkx2.5*, *Gata4*, *Tal1*), and 'blood' (*Tal1*, *Runx1*) -forming territories. In chick, *Tbx5* is expressed in FHF cells as they begin to differentiate into cardiomyocytes (*Nathan et al., 2008*; *Plageman and Yutzey, 2004*; *Tirosh-Finkel et al., 2006*), while *Nkx2.5* and *Isl1* are expressed throughout the anterior LPM in both FHF and SHF progenitors and pharyngeal endoderm (*Nathan et al., 2008*; *Tirosh-Finkel et al., 2006*). *Gata4* is expressed more lateral and posterior to these two factors. The endothelial/blood marker *Tal1* is highly expressed in the poste-

rior LPM and extraembryonic tissues, whereas the blood/hematopoietic lineage marker *Runx1* marks only extraembryonic blood islands, more lateral to cells expressing *Tal1*. This ISH analysis reveals the existence of both overlapping and distinct developmental fields (illustrated in *Figure 7—figure supplement 1*).

*Nkx2.5* mRNA expression at St. 7 is restricted to the cardiac crescent, while the enhancer is also expressed in the posterior LPM (*Figures 2*, *3* and *7*). Enhancer activity in the LPM could be explained by the lack of negative *cis* regulatory elements within this fragment. Alternatively, it could reflect the stability and perdurance of the fluorescent reporters (GFP or RFP), which would serve as lineage tracing tools marking all cells that have expressed *Nkx2.5* in their recent history. To address this issue, we performed RT-PCR analysis to ask whether *Nkx2.5* is expressed in early gastrulating St.3–4 embryos, when early cardiovascular progenitors emerge (*López-Sánchez and García-Martínez, 2011*). At these early stages cardiac progenitors were localized at the anterior primitive streak, while blood progenitors and *Nkx2-5*-en[+] were restricted to the mid-posterior streak (*Schoenwolf et al., 1992*; *Schultheiss et al., 1995*). Analysis of mid-

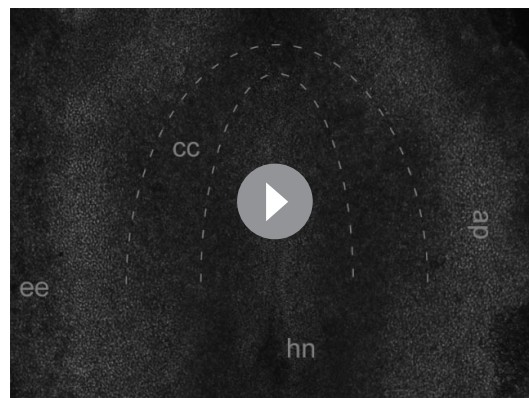

**Video 3.** Time lapse movie of a developing embryo expressing the Hb enhancer. Relates to *Figure 4*. A St.3 embryo electroporated with the Hb-en plasmid (GFP) and cultured overnight. The movie covers the development of the embryo between St.6 to St. 11–12. The BF and GFP channels were merged to better assess the migration of cells towards the inflow of the heart. Hb: hemangioblast; ee: extraembryonic; cc: cardiac crescent; nt: neural tube; hn; hensen's node; ap: area pellucida; da: dorsal aorta; oft: outflow; inf: inflow.

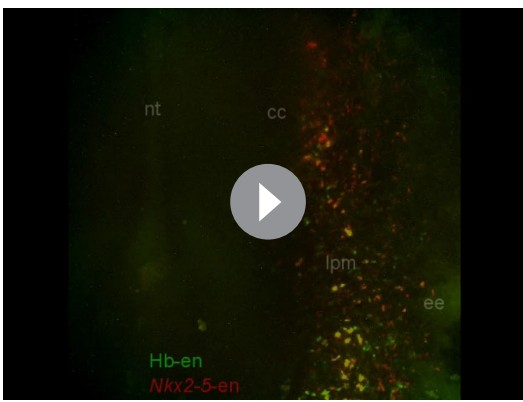

**Video 4.** Time lapse movie of a developing embryo co-expressing the Hb and *Nkx2-5* enhancers. Relates to *Figure 4*. A St.3 embryo electroporated with both the Hb-en (GFP) and *Nkx2-5*-en (RFP) plasmids subsequently cultured overnight. The movie covers the development of the embryo between St.6 to St.11. The GFP and RFP channels were merged to visualize the migration of Hb-en[+]/Nkx2-5-en[+] cells towards the inflow and the yolk sac.

primitive streak tissue detected *Nkx2.5*, but not *Isl1*, RNA expression, along with the earliest markers of mesoderm, *Mesp2* (homolog of mouse *Mesp1*) (*Saga et al., 1999*) and *Brachyury* (*Bry*) (*Figure 7B*; red and blue boxes, n = 18). Hemoangiogenic markers *Ets1*, *Flk1*, *Gata4*, *CD31*, Runx1 and *Tal1*, were also detected at these stages. Consistent with these finding, we detected expression of *Nkx2-5*-en in the mid-streak at these same stages, marking nascent mesodermal progenitors. Wholemount immuno-fluorescence staining of St. 4 embryos revealed multiple Nkx2.5[+] cells in and outside the primitive streak region (*Figure 7D*), while at St. 8 Nkx2.5 was expressed exclusively in the cardiac crescent (*Figure 7E*). *Nkx2.5* mRNA was also transiently expressed in the posterior LPM at St. 5–6 (*Figure 7F*, yellow box) in addition to strong and sustained expression within the anterior LPM (cardiac crescent). Lower expression was observed in extraembryonic tissue at St. 6, which was downregulated when embryos reached St. 7, in-line with inhibition of the cardiac developmental program in the posterior LPM and extra-embryonic tissue (*Schoenebeck et al., 2007*; *Van Handel et al., 2012*).

In-depth RT-PCR analysis at St. 7 embryos was performed for the extraembryonic mesoderm, LPM, cardiac crescent and neural plate tissues (*Figure 7—figure supplement 1*, n = 20). A mixture of endothelial and blood gene expression signatures (*Flk1*, *Tal1*, *Ets1*, *CD31*, *Gata1*, *Runx1*) was observed in both extraembryonic and LPM tissues, while the cardiac crescent expressed *Nkx2.5*, *Isl1* and *Tbx5* (*Figure 7—figure supplement 1*). Taken together, our molecular analyses suggest that in the chick transient expression of *Nkx2.5* marks hemoangiogenic progenitors in extra-cardiac regions from early primitive streak stages through St 6, just prior to formation of the cardiac crescent. These results are consistent with *Nkx2-5*-en serving as a lineage-tracing tool for *Nkx2.5*-expressing meso-dermal cells in gastrula and immediate post-gastrula stages.

## Ectopic expression of Nkx2.5 induces hemangioblast gene expression

Our findings suggest that *Nkx2.5* marks hemoangiogenic progenitors in chick nascent mesoderm. To examine the competence of nacent mesoderm to respond to Nkx2.5 expression, we performed Nkx2.5 gain-of-function experiments by electroporating the pCIG-*Nkx2.5* construct at St. 3, with analysis at St. 7 (cardiac crescent stage, *Figure 8*). Strikingly, enforced expression of Nkx2.5 at St. 3 induced robust expression of hemoangiogenic genes, including *Tal1*, *Flk1*, *Ets1* and *Lmo2* (red arrows, n = 6/8; n = 8/11; n = 4/5; n = 5/7, respectively, *Figure 8A'–D'*). Ectopic expression of these genes was observed at sites of high Nkx2.5 expression (lower panels), indicating a cell autonomous effect. Furthermore, Nkx2.5 overexpression inhibited somite formation and *Pax3* expression, indicating strong lateralization of the paraxial mesoderm (*Figure 8E–E"*; n = 5/6).

In a similar fashion, we performed overexpression experiments with Tal1 (*Figure 8—figure supplements 1* and *2*). Tal1 overexpression suppressed cardiac genes, mostly within the caudal part of the cardiac crescent (*Figure 8—figure supplement 1*, n = 21/27). Furthermore, the cardiac crescent appeared smaller in Tal1 electroporated embryos. At later stages (St. 12–13), Tal1 expression resulted in cardiac looping defects and the appearance of edemas (*Figure 8—figure supplement 2A–C*, n = 14/17). Furthermore, we observed ectopic blood island formation in the posterior LPM, where *Nkx2-5*-en was also expressed (*Figure 8—figure supplement 2D–I*, n = 10/17). Collectively, our data indicate that in the chick, nascent mid-streak mesoderm transiently expresses *Nkx2.5* mRNA. Furthermore, it is competent to respond to Nkx2.5 over-expression through engagement of a hemoangiogenic program. Overexpression of Tal1, a definitive hemogenic transcription factor,

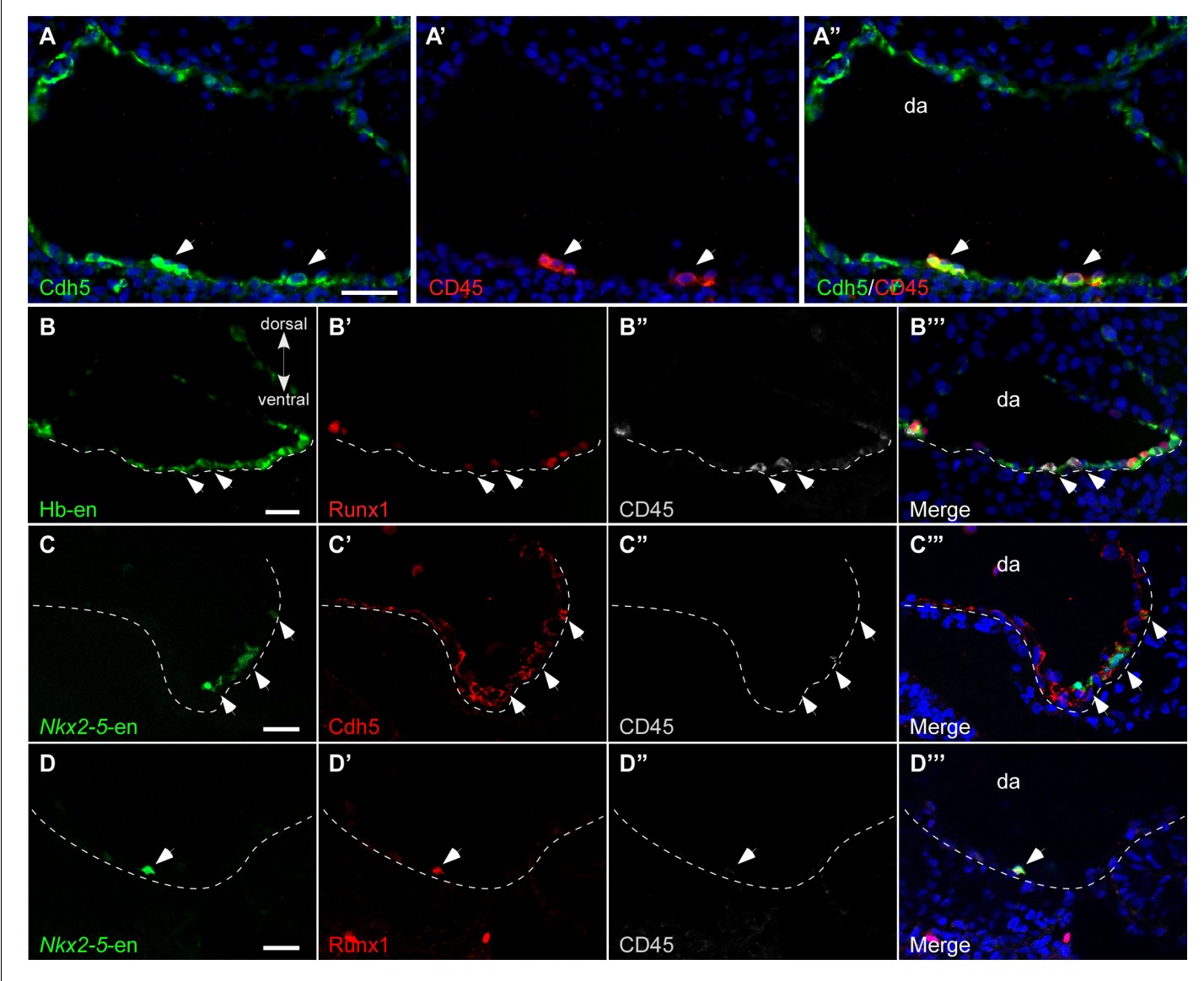

**Figure 5.** Hb-en[+] and *Nkx2-5*-en[+] cells contribute to the hemogenic endothelium of the dorsal aorta. (**A–A"**) A St.17 embryo sectioned at the level of the AGM and stained with novel anti Cdh5 (VE-cadherin) and CD45 antibodies delineating the endothelium and hematopoietic cells, respectively. (**B–B'''**) St.14–15 embryo electroporated with the Hb-en (GFP) and sectioned at the level of the early fused dorsal aorta stained for CD45, Runx1 and GFP. The arrow marks the initiation of hemogenic clusters indicated by CD45 expression. (**C**) Section of a St.14 embryo expressing the *Nkx2-5*-en in the right aspect of the newly fused dorsal aorta (arrowheads). While no CD45 expression is detected the *Nkx2-5*-en expression co-localized with Cdh5. (**D–D"**) Section of an early St.14 embryo expressing the *Nkx2-5*-en. Hemogenic endothelium cells are marked by Runx1 which precedes the expression of CD45. DA–dorsal aorta. Scale bars: 100 μm (**A**), 20 μm (**B–D**). Scale bar: 100 μm. n = 4/6. See also *Figure 4—source data 1*.

overides the endogenous cardiac program and redirects it to blood. Our data suggest that *Nkx2-5*-en provides a robust readout of Nkx2.5[+] lineage expansion and migration in the chick, and that the first role for Nkx2.5 in the embryo is as a hemoangiogenic transcription factor, before its latter involvement in cardiac regulatory networks.

## Mouse Nkx2-5 lineage[+] cells contribute to yolk sac vasculature

Previous CRE/loxP genetic lineage tracing studies in mouse embryos have demonstrated a contribution of Nkx2-5 lineage[+] cells to yolk sac blood island endothelium at E10.5 (*Stanley et al., 2002*) and hemogenic endocardium at E9.5 (*Nakano et al., 2013*). However, the origins of these cells and

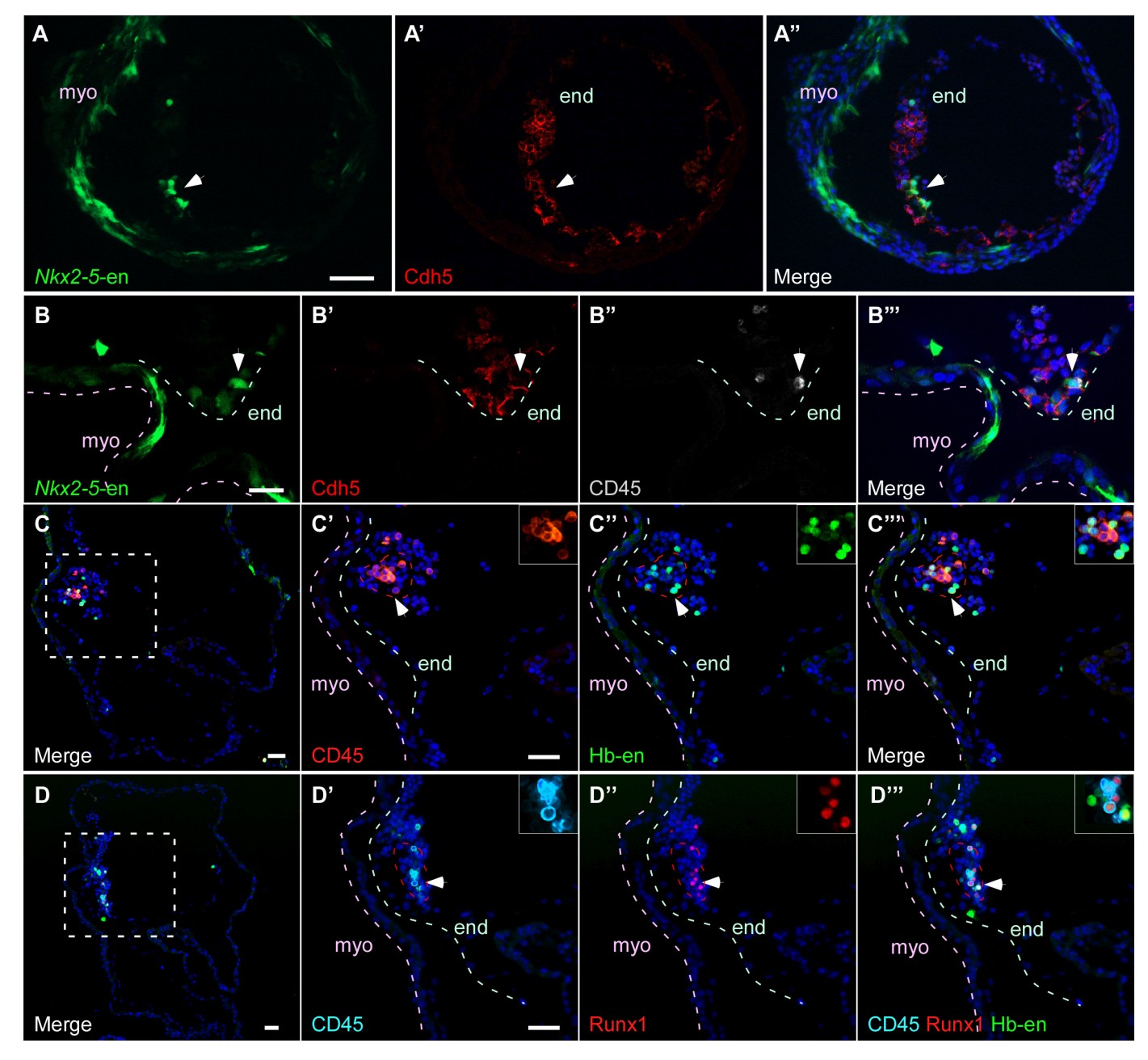

**Figure 6.** Hb-en[+] and *Nkx2-5*-en[+] cells give rise to hemogenic endocardium. (**A**) Section through a heart of a St.12–13 embryo expressing the *Nkx2-5*-en. Staining for Cdh5 delineates the endocardium (white arrow) which expresses the *Nkx2-5*-en. (**B**) High magnification of a section through the heart of a St13-14 embryo expressing *Nkx2-5*-en. A hemogenic cluster is attached to the endocardium delineated by Cdh5 and CD45 staining. (**C**) St.14–15 embryo electroporated with the Hb-en sectioned at the level of the heart. (**C'–C'''**) Higher magnification of the square box area marked in **C**. The section was stained for CD45 and shows a large hemogenic cluster in the endocardium. The arrow indicates Hb-en[+] cells that express CD45. The endocardium and myocardium are delineated by grey and pink dashed lines, respectively. The dashed red line delineates the hemogenic cell cluster. (**D**) St.14–15 embryo electroporated with the Hb-en and sectioned at the level of the heart tube. (**D'–D'''**) Higher magnification of the square box area marked in **D**. The section was stained for CD45, Runx1 and GFP. White arrow highlights an Hb-en[+] cell co-expressing CD45 and Runx1. For each dotted red line a higher magnification image is shown at the upper right side of the panel. end: endocardium; myo: myocardium. Scale bars: 100 μm, B – 20 μm. n = 4/6.

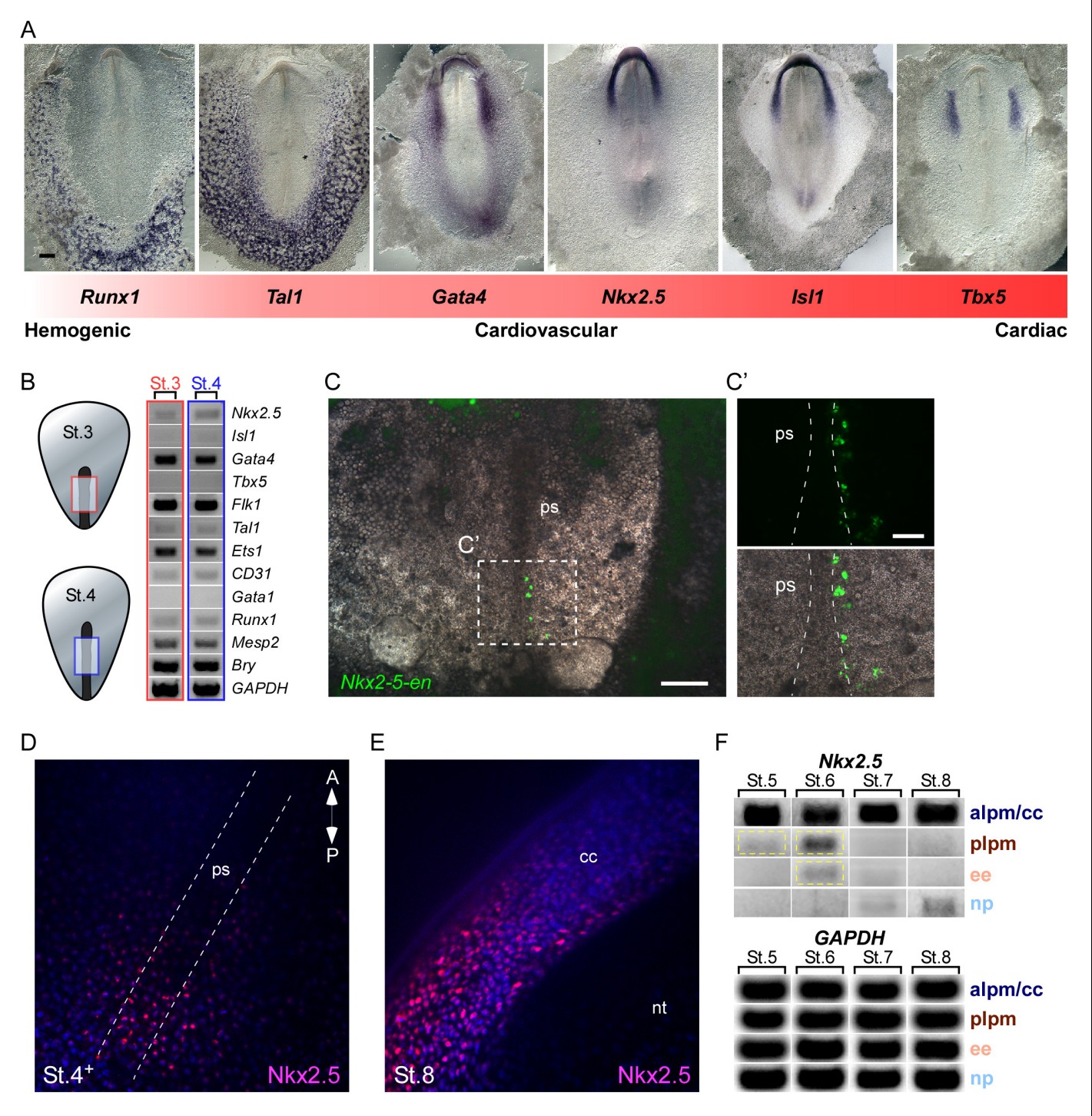

**Figure 7.** *Nkx2.5* is expressed in the nascent mesoderm in the chick primitive streak. (**A**) The cardiovascular network gradient. *In situ* hybridization of key genes involved in blood, vascular and cardiac development. (**B**) Semi qRT-PCR analysis of cardiovascular genes in early St. 3 (red box) and St. 4 (blue box) embryos. (**C–C'**) A St. 3–4 embryo electroporated with the *Nkx2-5*-en. (**C'**) shows a higher magnification of the square box in (**C**), displaying the GFP channel and the merge with the BF channel. (**D**) Wholemount immunofluorescence staining for Nkx2.5 on late St. 4 embryo. (**E**) Wholemount immunofluorescence of St. 8 embryo. Nkx2.5 expression is restricted to the cardiac crescent. (**F**) Semi qRT-PCR time course analysis of *Nkx2.5* mRNA expression at different embryonic stages and tissues. *Nkx2.5* expression is highlighted in yellow boxes. *GAPDH* is used as control. CC: cardiac crescent; NT: neural tube; PS: primitive streak. Scale Bars – **C**: 100 μm; **C'**: 25 μm.

The following figure supplement is available for figure 7:

*Figure 7 continued on next page*

*Figure 7 continued*

**Figure supplement 1.** Key cardiovascular gene expression gradient in a chick embryo.

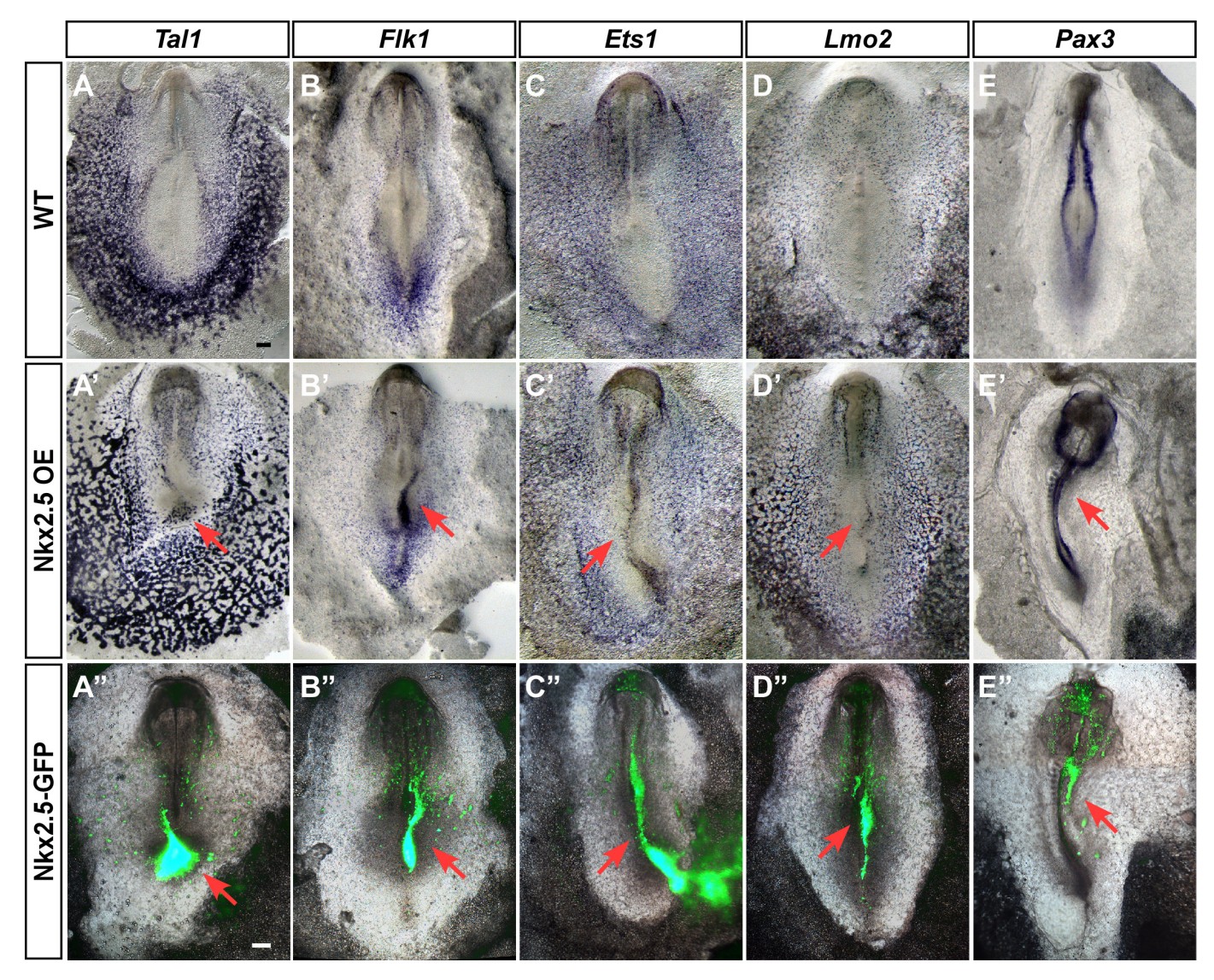

**Figure 8.** Ectopic expression of Nkx2.5 induces angioblast gene expression. (**A–E**) *In situ* hybridization for *Tal1*, *Flk1*, *Ets1* and *Lmo2* and *Pax3* in control embryos. (**A'–E'**) Embryos electroporated at St.3 with a vector over-expressing Nkx2.5 (pCIG-*Nkx2.5*-GFP). Embryos were subsequently incubated to St.7 and then subjected to *in situ* hybridization for the respected genes. Red arrow indicates where the ectopic expression of a gene was detected compared to its control embryo. (**A"–E"**) The same embryos as in **A'–E'** merged with the GFP channel visualizing the Nkx2.5 over-expressing tissues. The over-expressing phenotype corresponds to the area where the vector expression was the highest. Scale bar: 100 µm. See also ***Figure 9—figure supplements 1*** and ***2***.

The following figure supplements are available for figure 8:

**Figure supplement 1.** Tal1 over-expression inhibits cardiogenesis.

**Figure supplement 2.** Tal1 over-expression impairs normal heart development.

their contribution to the AGM region of the DA have not been explored. We crossed the *Nkx2-5$^{ires-cre}$* mice to *Rosa-LacZ, Rosa-YFP* or *Z/EG* reporter lines, and examined embryos for marked tissues at E7.5-E10.5. Whole mount staining for *β*-gal in *Nkx2-5$^{irescre/+}$;Rosa$^{lacZ/+}$* embryos revealed the presence of isolated labelled cells in extra-embryonic yolk sac mesoderm at ~E70.5-E8.0, coincident with the earliest first appearance of labelling in the cardiac crescent, although not in pre-crescent embryos (*Figure 9A*; *Figure 9—figure supplement 1A,A'*). More labelled yolk sac cells were evident during heart tube formation (*Figure 9B*). Labelled cells tended to be proximal to the crescent and restricted to the anterior half of the yolk sac (*Figure 9A,B*). As development progressed further, the number of labelled cells increased dramatically, such that by E9.0 there were many positive cells associated with the yolk sac vasculature (*Figure 9D*, *Figure 9—figure supplement 1D,D'*). At this stage, isolated lineage-tagged cells were also seen within the cardio-pharyngeal region of the embryo extending into the head, as well as more caudally in the trunk and tail (*Figure 9C*). Histological sections of yolk sac showed that from E8.0, lineage tagged cells contributed only to the mesodermal layer including the endothelial lining of forming vessels (*Figure 9—figure supplement 1B–D''*). Moreover, at E9.0, the majority of *Nkx2-5* lineage-traced cells in the yolk sac vasculature also expressed endothelial markers Vegfr2/Flk1 and VE-cadherin/Cdh5 (*Figure 9—figure supplements 2A–E'''*), suggesting their endothelial nature. However, the yolk sac vasculature was a mosaic of lineage labelled and unlabelled cells (*Figure 9—figure supplement 1D'*). Labelled blood cells were also evident within vessels (*Figure 9—figure supplement 1D,D''*). We note that Nakano and colleagues did not see significant labelling in yolk sac at E9.5 using identical (*Nkx2-5$^{irescre/+}$;Rosa26$^{lacZ/+}$*) or similar (*Nkx2-5$^{irescre/+}$;Rosa26$^{YFP/+}$*) lineage tracing crosses (*Nakano et al., 2013*), which we attribute to the lower sensitivities of CRE reporters on different genetic backgrounds.

To explore further the lineage identities of those cells first expressing *Nkx2-5* in mouse embryos at E7.5, we interrogated recently published transcriptome data of single cells derived from the epiblast and mesoderm of E7.0-E7.75 embryos, whose lineage identities had been determined based on cell surface marker status (Flk1; CD41) and transcriptome signature (*Scialdone et al., 2016*). No cells classified as epiblast or extra-embryonic ectoderm expressed *Nkx2-5* (*Figure 9—figure supplement 3A–C*). However, *Nkx2-5* was expressed in ~7% of cells classified as nascent mesoderm and ~30% of cells classified as posterior and pharyngeal mesoderm (likely cardiac progenitors). Importantly, ~30% of endothelial cells and an equivalent proportion of cells classified as allantois (one extra-embryonic site of hemopoiesis) were *Nkx2-5$^{+}$*, as were ~10% of early blood progenitors. These data confirm the hemoangiogenic nature of Nkx2-5 lineage-traced cells in early mouse embryos.

## Onset of Nkx2-5 expression in mouse embryos

It is noteworthy that Nkx2-5-Cre mediated lineage-tagged cells were not found in late gastrula/pre-crescent stage embryos at E7.0 (*Figure 9—figure supplement 1A,A'*). Lack of significant *Nkx2-5* mRNA expression in early to mid-gastrula stage embryos was supported by interrogation of a previously published high-resolution spatial RNA-seq map of E7.0 mid-gastrula stage embryos (*Peng et al., 2016*), as well as similar maps from E6.5 early gastrula and E7.5 late gastrula/pre-cardiac crescent stage embryos (unpublished data from S.S., J-D.J.H., G.P., N.J. and P.P.L.T.; summarized in *Figure 9—figure supplement 4A–E*). No consistent spatial pattern of *Nkx2-5* expression was obtained in proximo-distal and anterior-posterior embryonic segments of epiblast at early gastrulation (n = 4, *Figure 9—figure supplement 4A*). Where expression did register, it occurred in sporadic segments, and the mean and maximum normalized read counts (log$_{10}$ of fragments per kilobase million [FPKM]) were very low (mean 0.18 ± 0.08, range 0.09–0.25; *Figure 9—figure supplement 4D,E*). Similarly, at mid-gastrulation (E7.0; n = 3), only very low read counts registered in epiblast (mostly in a single replicate; mean 0.07 ± 0.07, range 0.032–0.32), and nascent mesoderm (mean 0.58 ± 0.67, range 0.09–1.35) (*Figure 9—figure supplement 4B,D,E*). In contrast, in E7.5 late gastrula/pre-cardiac crescent stage embryos, significantly higher read counts registered in coherent clusters of segments in nascent mesoderm in 2 of the three replicates (mean 5.6 ± 5.9, range 0.07–20.1) while only low counts were found in ectoderm in sporadic segments (mean 0.32 ± 0.36, range 0.03–1.1) (*Figure 9—figure supplement 4D,E*). The higher number of counts in clusters of segments in E7.5 nascent mesoderm likely reflects the upward slope of the Nkx2-5 expression profile during specification of cardiomyocytes in the cardiac crescent.

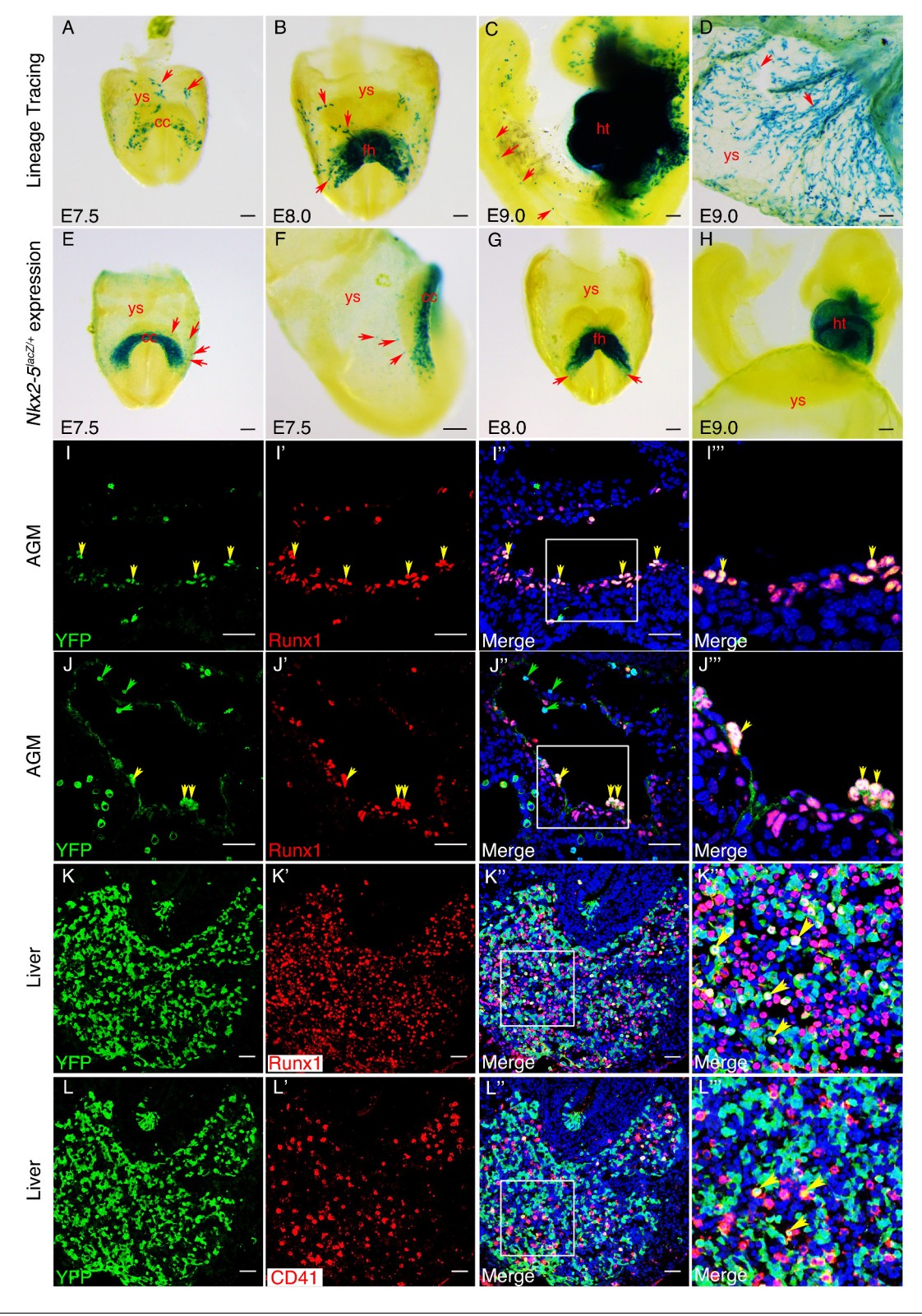

**Figure 9.** An *Nkx2-5+* lineage contributes to hemogenic endothelium of the dorsal aorta in the mouse. (A–D) Lineage tracing β-gal staining of *Nkx2-5iresCre;R26R* embryos. (A) E7.5 showing β-gal staining beginning in the cardiac crescent (cc) and yolk sac (ys). (B) E8.0 showing β-gal staining in the forming heart (fh) and yolk sac. Red arrows in A and B indicate β-gal positive cells in the anterior half of yolk sac. (C) E9.5 showing β-gal staining in the looped heart (ht) and associated pharyngeal region extending into the head (potentially angiogenic cells or blood cells). Isolated cells are also found

*Figure 9 continued on next page*

*Figure 9 continued*

within the trunk and tail regions of the embryo (red arrows). (D) E9.5 showing β-gal staining in the yolk sac marking angiogenic and potentially hemoangiogenic cells within the vasculature (red arrows; see sections in *Figure 9—figure supplement 1B–D'*). (E–H) Nkx2-5 expression β-gal staining of *Nkx2-5^lacZ/+* embryos. (E–F) E7.5 showing *Nkx2-5*-LacZ expression in the cardiac crescent and proximal yolk sac. Red arrows show single β-gal positive cells in yolk sac in close proximity to the cardiac crescent. (G–H) E8.0 (G) and E9.0 (H) embryos showing *Nkx2-5*-LacZ expression in the forming heart and looped heart respectively. Red arrows show caudal parts of the cardiac zone that contain positive cells in yolk sac. (I–I''') E10.5 *Nkx2-5^irescre*; *ROSA^YFP* embryos sectioned at the level of the aorta gonad mesonephros (AGM) region and immunostained for Runx1 (red) and YFP (green). Co-labelled Runx1⁺ YFP⁺ cells are present mostly in the floor of the dorsal aorta (da). (J–J''') E10.5 *Nkx2-5^irescre;ROSA^YFP* embryo depicting *Nkx2-5* lineage⁺ (YFP⁺) hemogenic cells (Runx1⁺) protruding from the endothelial layer of the dorsal aorta in the AGM region. (K–L''') E10.5 embryos showing co-staining for *Nkx2-5*-lineage⁺ (YFP⁺) and Runx1⁺ (K–K''') or CD41⁺ (L–L''') cells in the developing liver. Yellow arrows in I-K''' mark YFP⁺ Runx1⁺ cells and in L-L''' mark YFP⁺ CD41⁺ co-labelled cells. Green arrows in J and J'' depict YFP⁺ Runx1⁻ single positive cells. Scale bar: **A-H** - 100 μm; **I-J''** - 50 μm; **K-L''** - 100 μm.

The following figure supplements are available for figure 9:

**Figure supplement 1.** *Nkx2-5* is transiently expressed in yolk sac mesoderm from cardiac crescent stages.

**Figure supplement 2.** *Nkx2-5*⁺ lineage contributes to hemogenic endothelium of yolk sac vasculature.

**Figure supplement 3.** *Nkx2-5* expression in single cells of mouse E7. 0–7.75 embryos.

**Figure supplement 4.** Spatial RNA-seq analysis of Nkx2-5 expression in mouse gastrula-stage embryos.

**Figure supplement 5.** *Nkx2-5* lineage⁺ cells in the cardio-pharyngeal region.

The RNA-seq profiles in mouse embryos are therefore consistent with our lineage tracing data detecting the onset of *Nkx2-5* expression at the pre-crescent/crescent transition. Taking a complementary approach, we explored the timing and location of *Nkx2-5* expression using the sensitive *Nkx2-5^LacZ* knockin allele (*Prall et al., 2007*)(*Figure 9E–H*). Similar to our findings with lineage tracing and RNA-seq, the earliest expression of β-gal in extra-embryonic tissue of *Nkx2-5^LacZ/+* embryos was at E7.5, when the cardiac crescent was established (*Figure 9E,F*; *Figure 9—figure supplement 1E–F*). Even when only few cells were β-gal positive in the crescent, a comparable number of positive cells were evident in the yolk sac (*Figure 9—figure supplement 1F*). Labelling was in the mesodermal layer proximal to the cardiac crescent (*Figure 9E,F*; *Figure 9—figure supplement 1F–G'*). However, by forming heart tube and looping heart stages, positive cells in the yolk sac were only rarely evident (*Figure 9G,H*), although visual inspection of whole mount stained E8.0 embryos and histological sectioning, showed positive cells in yolk sac at the caudal territories of the cardiac zone (*Figure 9G*, arrows) and in yolk sac mesoderm and endoderm immediately overlying the heart region (*Figure 9—figure supplement 1H,H'*). Interestingly, both lineage tracing and *Nkx2-5*-LacZ expression showed that labelled endodermal cells were not seen in the yolk sac layer distal to the cardiac zone, and those located proximally likely remain associated with the foregut. We conclude that in mouse, *Nkx2-5* is activated transiently in extra-embryonic mesoderm in close association with the cardiac crescent at E7.5-E8.0. The combination of expression analysis, lineage tracing and RNA-seq suggests that these cells expand as they move deeper into the yolk sac and populate vascular structures. We anticipate that the Nkx2-5⁺ cells present at E7.5-E8.0 are the founders of the Nkx2-5 extra-cardiac lineage and are analogous to those detected in extra-embryonic regions in chick embryos. We hypothesize that some Nkx2-5⁺ cells correspond to hemoangiogenic precursors that later populate the endothelium of yolk sac blood islands and hemogenic endocardium (our study and *Nakano et al., 2013*; *Stanley et al., 2002*).

## Mouse Nkx2-5 lineage⁺ cells contribute to the hemogenic endothelium of the dorsal aorta

We next explored whether the mouse Nkx2-5 extra-embryonic lineage⁺ cells described above also contribute to the hemogenic endothelium of the AGM region in the DA. Nakano and colleagues did not detect Nkx2.5 lineage⁺ cells in the AGM at E9.5 using the crosses mentioned above

(*Nakano et al., 2013*). However, in *Nkx2-5$^{irescre/+}$;Rosa$^{YFP/+}$* embryos at E10.5, we found abundant YFP$^+$ cells in the AGM region, including in endothelium and sub-endothelium (*Figure 9I–J'''*; *Figure 9—figure supplement 1J–J'''*), as seen previously for cells in the AGM expressing the earliest hemogenic marker Runx1 (*North et al., 2002*). We also saw cells with the same expression profile within the liver (*Figure 9K–K'''*), the first site of definitive mammalian embryonic hemopoiesis. Most YFP$^+$ cells in the DA co-expressed Runx1, although there were also many Runx1$^+$ YFP$^-$ cells in the region (*Figure 9I–J'''*; *Figure 9—figure supplement 1I*). Quantifications showed that ~ 15% of Runx1$^+$ cells in the endothelium and ~16% of Runx1$^+$ cells in the sub-endothelium were also YFP$^+$, while overall >77% of YFP$^+$ cells were also Runx1$^+$ (*Figure 9—figure supplement 1I*). Consistently, we calculated a similar ~15% ratio of Runx1$^+$/*Nkx2-5*-en$^+$ in the chick DA (data not shown). Clusters of Runx1$^+$ YFP$^+$ cells appeared to be budding from the endothelial layer of the DA towards the lumen (*Figure 9J–J'''*, *Figure 9—figure supplement 1J–J'''*), behaviour consistent with that of hemogenic endothelium (*Jaffredo et al., 1998*). With a single exception, none of the YFP$^+$ cells in the AGM expressed the hemogenic marker CD41 (integrin alpha2b) that is known to act downstream of Runx1 (*Mikkola et al., 2003*). However, there were significant numbers of YFP$^+$ CD41$^+$ as well as YFP$^+$ Runx1$^+$ cells in the liver (*Figure Figure 9K-L'''*).

In line with previous mouse data (*Nakano et al., 2013*; *Paffett-Lugassy et al., 2013*; *Stanley et al., 2002*) and chick lineage tracing studies presented above, mouse Nkx2-5 lineage$^+$ cells were detected within the endothelium of the pharyngeal arch arteries and paired dorsal aorta, and endocardium of the atria and ventricles, at E9.5 (data not shown) and E10.5 (*Figure 9—figure supplement 5A–E''*). Using *Nkx2-5$^{irescre/+}$;Z/EG* embryos, we confirmed that Nkx2-5 lineage$^+$ (GFP$^+$) cells were embedded within Pecam1$^+$ endothelium of the DA (*Figure 9—figure supplement 5E–E''*). Some Nkx2-5 lineage$^+$ endothelial cells in the cardiac region at E10.5 were also CD41$^+$, although these were very rare (*Figure 9—figure supplement 5C–D*).

## Discussion

The present study provides novel insights into early cardiogenesis and hemangiogenesis in vertebrate embryos. First, we revealed an early contribution of mesodermal cells residing outside the classical cardiac crescent region, to the developing heart (*Figure 10*). Cellular and molecular analyses indicate that a subset of extra-embryonic/LPM-derived hemogenic angioblasts travels via the inflow tract to populate the heart endocardium. Moreover, our results in both chick and mouse models demonstrate that the hemogenic endothelium in the DA and endocardium derives, at least in part, from Nkx2.5 lineage$^+$ angioblast progenitors that may also contribute to hematopoietic stem cells (HSCs) in the fetal liver (*Figure 10*). Based on these studies and gain of function experiments, we propose a broader role for Nkx2.5 in the cardiovascular and blood lineage diversification programs than currently appreciated.

The cardiovascular and hematopoietic lineage specification programs begin at gastrulation, when nascent mesodermal cells ingress through the primitive streak. In the chick, we demonstrated that Nkx2.5 is transiently expressed in the mid-primitive streak and the posterior LPM/extra-embryonic tissues at St. 4–6. Moreover, overexpression of Nkx2.5 induced endothelial/hematopoietic gene expression. In the mouse, *Nkx2-5* activation in extra-embryonic tissue occurred later in gastrulation than in the chick, at the time of cardiac crescent formation, but positive cells nonetheless contributed extensively to the yolk sac vasculature and blood. Expression was also seen in hemogenic endothelium of the AGM, although we have not yet shown definitively in mouse whether positive cells migrate to the AGM from extra-embryonic tissue or whether Nkx2.5 is expressed de novo in hemogenic angioblasts arising in embryonic LPM. Confirming findings of Nakano and colleagues (*Nakano et al., 2013*), we found Nkx2.5 lineage$^+$ blood cells in the yolk sac and embryonic vessels. Nakano et al. found that Nkx2.5 lineage$^+$ blood cells expressed haemoglobins typical of primitive then definitive erythropoiesis, although no positive blood cells were present in adults. These results resonate with those of Yoder et al., who demonstrated that the yolk sac contains HSCs capable of sustaining long-term multi-lineage hemopoiesis when transferred to conditioned hemopoietic organs of neonates although not adults (*Yoder et al., 1997*). Taken together our findings suggest that Nkx2.5 plays a conserved role in the establishment of the cardiovascular and early blood cell lineages, in apparent agreement with the severe defects observed in the yolk sac vasculature and hematopoiesis seen in *Nkx2-5* mutant mouse embryos (*Tanaka et al., 1999*).

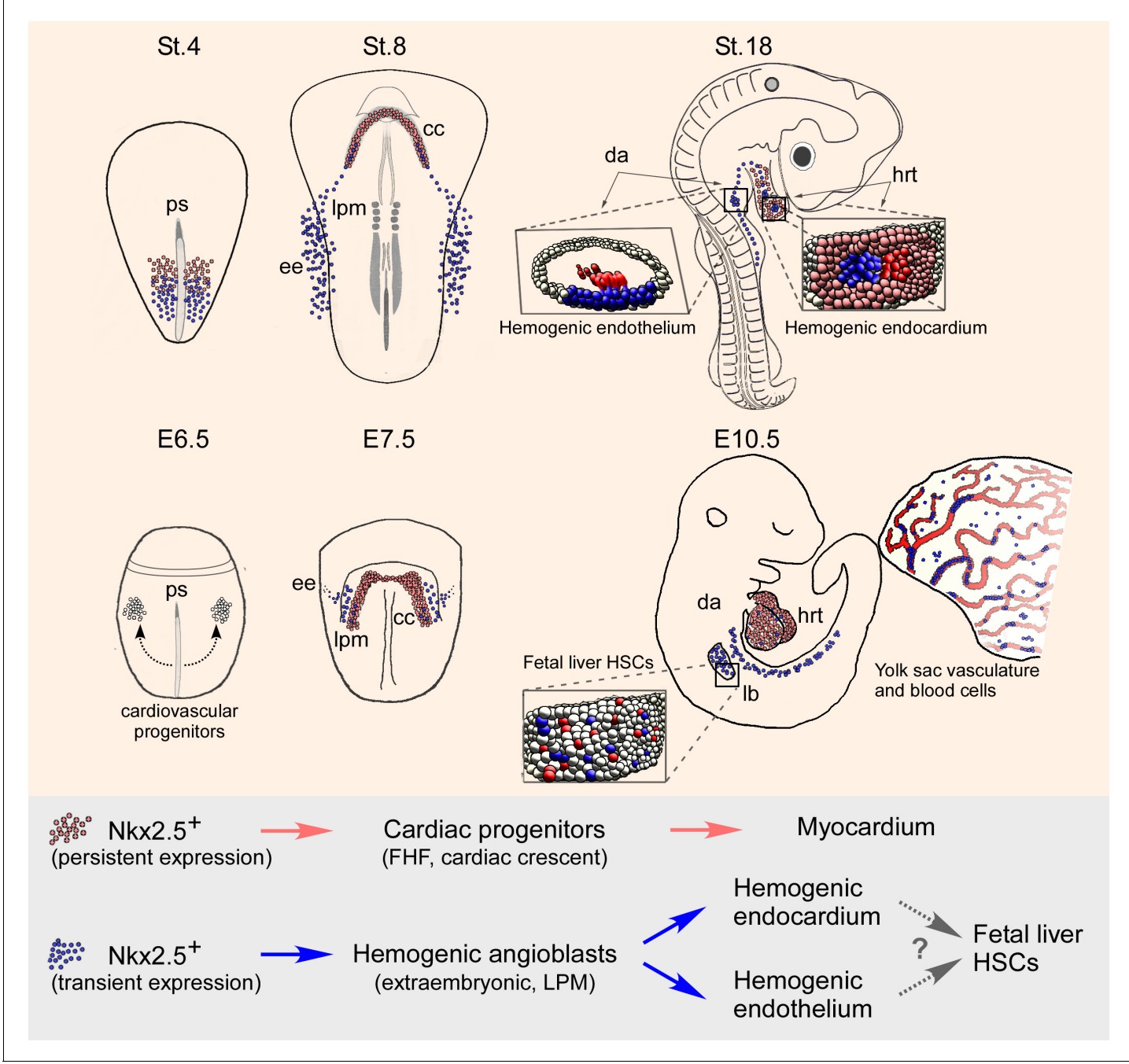

**Figure 10.** *Nkx2.5* marks hemogenic angioblasts that contribute to the formation of the endocardium and dorsal aorta. Cardiovascular progenitors begin to form at the onset of gastrulaion. These populations segregate as the cells begin to migrate in a lateral fashion towards the extraembryonic tissue. During early gastrulation in the chick (St.4) or late gastrula stages in the mouse (cardiac crescent, E7.5) Nkx2.5+ hemogenic angioblasts are specified. While Nkx2.5 expression is maintained at the cardiac crescent its expression in the hemogenic angioblasts is downregulated. *Nkx2.5* lineage-derived cells populate the cardiac crescent, extraembryonic tissue and lateral plate mesoderm in both species (St.8 and E7.5). As the embryo develops *Nkx2.5* derived hemogenic angioblasts migrate to the endocardium and dorsal aorta, there generating blood cells via the hemogenic endothelium (St.18 and E10.5). In both chick and mouse *Nkx2.5* lineage derived cells contribute massively to the yolk sac vasculature and to HSCs in the fetal liver (mouse). CC – cardiac crescent; DA – dorsal aorta; EE – extraembryonic; FHF – first heart field; hrt – heart; LB – liver bud; LPM – lateral plate mesoderm; PS – primitive streak.

The electroporation technique in early chick embryos enabled us to follow nascent mesoderm progenitors with hemogenic angioblastic characteristics. What makes these cells hemangioblast-like? First, these extra-embryonic/LPM mesoderm progenitors were labelled by both *Nkx2-5*-en and hemangioblast enhancers. Morphologically, we observed the expression of these two enhancers within blood islands, and could track their migration into the DA (and other vessels), as well as to the endocardium. These cells broadly expressed hematopoietic and endothelial lineage markers such as *Runx1*, *Flk1*, *Tal1*, *Ets1*, *Gata1*, *Lmo2* and *Gata4*. When cultured *in vitro*, these cells differentiated into endothelial and blood progenitors, although we have not yet shown a bipotent differentiation potential in a clonal assay. Finally, we showed that in vivo, these cells contributed to the hemogenic endothelium in the DA and heart endocardium.

Tight molecular and cellular relationships exist between myocardial, endocardial, endothelial, and hematopoietic lineages in different organisms. A number of transcription factors control these cell fate choices, primarily through cross-inhibition mechanisms (*Caprioli et al., 2011*; *Schoenebeck et al., 2007*; *Van Handel et al., 2012*). Heart, vessels and blood likely represent an ancient lineage triad controlled by Nkx2-5 that has contributed to cardiovascular and hemogenic developmental plasticity in vertebrates. This plasticity is highlighted by activation of the hematopoietic program within hemogenic endothelium at various embryonic sites, including the yolk sac (*Palis et al., 1999*), DA (*Jaffredo et al., 1998*), extra-embryonic arteries (*Gordon-Keylock et al., 2013*), and endocardium (*Nakano et al., 2013*).

The mouse enhancer for *Nkx2-5* labels a broad cardiovascular mesoderm population in the chick, including FHF cardiac progenitors, endothelial and hematopoietic progenitors. Because of the demonstrated early expression of Nkx2.5 protein and mRNA in the nascent mesoderm at St. 4–6, we suggest that the reporter expression provides a valid readout of the transient expression of *Nkx2.5* in these cells, which is later downregulated in extra-cardiac lineages by endothelial and hematopoietic transcription factors. In the mouse, we showed that *Nkx2-5* lineage[+] cells arise in extra-embryonic tissues later in gastrulation associated with cardiac crescent formation. Linage[+] cells also contributed substantially to the hemogenic endothelium within the AGM, liver, and, to a lesser degree, the endocardium. The extra-cardiac Nkx2-5 lineage[+] cells that contribute to mouse hemopoiesis in the embryo from E8.5-E15.5 (*Nakano et al., 2013*), likely derive from the yolk sac or AGM region via the liver, rather than the endocardium as proposed (*Nakano et al., 2013*). However, whether there is a contribution of Nkx2-5 lineage[+] cells to primitive hemopoiesis in the mouse yolk sac, and how Nkx2-5 lineage[+] cell arrive in the AGM region, remain to be explored. Moreover, how Nkx2-5 lineage[+] cells relate to the Runx1[+] Gata1[-] Etv2[+] VE-cadherin[+] hemogenic angioblasts which are already present in the yolk sac at E7.5 (*Eliades et al., 2016*; *Lie-A-Ling et al., 2014*; *Tanaka et al., 2014*), needs detailed analysis. Notwithstanding the differences in the timing of activation of Nkx2-5 in extraembryonic tissue in chick and mouse, our findings on the Nkx2-5 lineage in these two models seem highly concordant.

Bioinformatics and *in vitro* analyses demonstrated that several *Gata* and *Ets* sites are present and active within the *Nkx2-5* enhancer (data not shown and *Lien et al., 1999*). Our ISH and RT-PCR analysis revealed the expression of *Gata4* in both LPM and cardiac crescent (and also nascent mesoderm) corresponding to the expression of the *Nkx2-5*-en. Hence, Gata factors could drive the early expression of *Nkx2.5*. Scl/Tal1 was previously shown to be a key regulator of hemangioblasts (*Bussmann et al., 2007*; *Schoenebeck et al., 2007*; *Van Handel et al., 2012*). Our findings demonstrating the ectopic activation of *Tal1* by Nkx2.5 suggest that the latter acts upstream of Tal1 in the hemangioblast hierarchy and may have a role in priming hemopoiesis. Activation of Tal1 in the chick inhibited cardiac gene expression, in line with the repressive role of Scl/Tal1 on cardiogenesis, as demonstrated in Scl/Tal1 mutant mouse embryos in which ectopic cardiomyogenesis was observed in the yolk sac vasculature and endocardium (*Van Handel et al., 2012*). Similar to Scl/Tal1, loss of etsrp/etv2 induced cardiac gene expression in endocardial progenitors (*Palencia-Desai et al., 2011*; *Rasmussen et al., 2011*). Taken together, these studies and our data highlight the dynamic developmental interplay between embryonic mesoderm progenitors for cardiac, endothelial and blood lineages.

Endocardial cells are known to be heterogeneous in origins (*Harris and Black, 2010*; *Milgrom-Hoffman et al., 2011*; *Vincent and Buckingham, 2010*). *In vitro* studies in chick and mouse provided evidence for bipotential progenitors for myocardial and endocardial cells that exist in the SHF (*Hutson et al., 2010*; *Kattman et al., 2006*; *Lescroart et al., 2014*; *Moretti et al., 2006*), but

probably not in the FHF in which myocardial progenitors are unipotent (*Lescroart et al., 2014*; *Später et al., 2013*). Our findings are consistent with the idea that endocardial and myocardial lineage separation occurs very early during embryogenesis, in line with other studies (*Li et al., 2015*; *Milgrom-Hoffman et al., 2011*; *Paffett-Lugassy et al., 2013*).

Understanding the origin and regulation of definitive HSCs is a subject of great interest and contention. It is clear that HSCs are mesoderm-derived and their genesis involves differentiation from endothelial cells. In the chick, zebrafish and mouse embryos, the splanchnic mesoderm has been considered to be the source of hemogenic endothelium-derived HSCs (*Childs et al., 2002*; *Lancrin et al., 2009*; *Zovein et al., 2010*). More recent data suggest that hemogenic endothelium at embryonic sites have their origins in the yolk sac (*Tanaka et al., 2014*). It is likely that multiple sources of endothelial cells contribute to the formation and establishment of HSCs including extra-embryonic hemogenic angioblasts (*Tanaka et al., 2014*) and intra-embryonic angioblasts that are induced to form HSCs by local signalling (*Nguyen et al., 2014*; *Richard et al., 2013*). Consistent with these findings we suggest that part of the hemogenic endothelium in both DA and endocardium derives from extra-embryonic/LPM hemogenic angioblast progenitors previously expressing Nkx2.5. The detection of Nkx2.5 lineage[+] hematopoietic progenitors in DA, liver and endocardium, suggests that these progenitors derive from a common embryonic origin, although this remains to be formally tested. Potentially relevant to this work, a lineage tracing study with *Nfatc1-Cre* mice suggests that endocardial cells, which are in contact with liver epithelium, contribute to the liver vasculature (*Zhang et al., 2016*).

As in the DA, endocardial cells have multiple origins, some of which derive from specialized angioblasts with hemogenic properties. Indeed, the existence of specialized angioblasts has now been documented in the floor of the cardinal vein that generates lymph, arterial and venous cell fates (*Nicenboim et al., 2015*). Thus, analysis of heart development has revealed on the one hand major contributions from the well-characterised cardiac cell populations (e.g. FHF, SHF, cardiac neural crest, epicardium), along with minor contributions from other mesodermal progenitors that provide further developmental plasticity and functionality.

Our studies and those of others (see Introduction) suggest a conserved role for Nkx2.5 in both heart and hemoangiogenic lineage development. In addition to its role in the specification of cardiac mesoderm, the *Drosophila* Nkx2.5 homologue, tinman, is essential for formation of the fly larval lymph gland, a hemopoietic organ proposed to be similar to the hemogenic endothelium of the mammalian AGM region (*Mandal et al., 2004*). Thus, cardio-hemoangiogenic lineage development may be one of the ancestral functions of Nkx2.5. In summary, the role of Nkx2.5 in endothelial and blood specification programs is evolutionarily conserved from fly to human. In vertebrates, Nkx2.5 expression is mostly restricted to cardiac progenitors, but in a narrow window of time may prime vascular and blood development, as shown already in pharyngeal arch arteries and endocardium (*Nakano et al., 2013*; *Paffett-Lugassy et al., 2013*). The definitive vasculature and blood networks ultimately utilise other transcription factors for lineage specialisation, which also prevent ectopic heart formation in these highly plastic mesodermal cells (*Van Handel et al., 2012*). Unravelling the dynamic roles of Nkx2.5 in cardiac, endothelial, and blood lineage specification will lead to a deeper understanding of the cardiovascular/blood regulatory networks, and may point the way toward novel regenerative therapies for cardiovascular disease.

## Materials and methods

### Eggs and embryos

Fertilized white eggs from commercial sources were incubated to the desired stage at 38.0°C in a humidified incubator, according to Hamburger and Hamilton (*Hamburger and Hamilton, 1951*).

### Mouse lines

Mice used in this study were bred and maintained in the Victor Chang Cardiac Research Institute BioCore facility according to the Australian Code of Practice for the Care and Use of Animals for Scientific Purposes. For the *Nkx2-5*-lineage study, the *Nkx2-5^irescre* strain (*Stanley et al., 2002*) was crossed with *Rosa^lacZ* (*Soriano, 1999*), *Rosa^YFP* (*Srinivas et al., 2001*) or *Z/EG* (*Novak et al., 2000*)

reporter lines. Litters of required embryonic stages were dissected in 1X ice cold PBS, fixed and processed accordingly.

## Whole mount β-Gal staining

For whole mount $\beta$-gal staining, embryos were fixed in 0.1 M sodium phosphate buffer containing 0.2% glutaraldehyde, 1.48% formaldehyde, 5 mM EGTA and 2 mM magnesium chloride, pH7.3 from 30 min to 60 min at room temperature depending on the size of the embryo (for examples, E9.0 embryos were fixed for 1 hr). Further, embryos were washed in 0.1M sodium phosphate buffer containing 2 mM of magnesium chloride, 0.01% sodium deoxycholate and 0.02% of Nonidet-P40 detergent followed by $\beta$-gal staining in 0.1M sodium phosphate buffer containing 1X potassium ferrocynide, 1X potassium ferricynide and 1 mg/ml of X-gal (Promega, V3941). Imaging of whole mount beta-gal stained embryos was done with Leica M125 microscope fitted with a Leica DFC295 camera.

## Mouse sectioning and immunostaining

Embryos were fixed in 4% PFA at 4° C overnight followed by PBS washes. Subsequently, embryos were transferred through a 15% and 30% sucrose gradation followed by embedding in Tissue-Tek O.C.T. compound (Thermo scientific, Waltham, MA USA). 10 µM sections were prepared using Leica cryostat. For immune-staining sections were fixed in 4% PFA for 5 min on ice, washed in a 1X PBS, blocked in serum containing 3% goat serum, 3% bovine serum albumin and 0.1% Triton X-100 followed by incubation with primary antibodies overnight at 4° C. Primary antibodies used were GFP (1:200, ab13970), Runx1 (1:200, ab92336), CD41 (1:300, ab33661), CD31 (1:100, 553370, BD Pharmingen), Vegfr2/Flk1 (1:100, sc48161) and VE-cadherin (1:100, sc-6458). Secondary antibodies, anti-rabbit biotin, anti-rat biotin, anti-goat biotin and anti-chicken Alexa Fluor-488 (Life Technology) were used at 1:200 concentrations. For biotinylated antibodies signal was amplified using ABC (Vectastatin) and cy3-Tyramide amplification kit (Perkin Elmer). Images were taken using a Zeiss LSM 700 Upright confocal microscope.

## Quantification of Nkx2-5 lineage cells in mouse AGM

Runx1$^+$ YFP$^-$, Runx1$^-$ YFP$^+$ and Runx1$^+$ YFP$^+$ cells were quantified in endothelium or sub-endothelium of the DA in the region of AGM from three different embryos. Confocal Z-stack images from six to eight 10 µm sections per embryo were used for counting. The average number of single or double positive cells per section were graphed with standard deviation.

## Whole mount in situ hybridization

Whole mount in situ hybridization was performed, using digoxigenin (dig)-labelled antisense riboprobes synthesized from total cDNA. Briefly riboprobes were generated using T7 polymerase in the presence of dig-labelled UTP (Roche, Penzberg, Germany). Embryos were incubated to the desired stage, and fixed in 4% paraformaldehyde (PFA) overnight. Dehydration was performed with methanol series, after which the embryos were stored in absolute methanol overnight at −20℃. Rehydration was performed in methanol series, in PBT. Embryos were treated with proteinase K and fixed with 4% PFA/0.2% glutaraldehyde. Prehybridization (50% formamide, 5XSSC pH 4.5, 2% SDS, 250 µg/ml yeast tRNA, 50 µg/ml heparin) was performed at 68℃, prior to hybridization overnight at that temperature. To remove unbound probe, a series of washes (50% formamide, 5XSSC pH 4.5, 1% SDS) was performed. After a second series of washes in 0.1M maleic acid (Ph 7.4), 0.15M NaCl, and 1% Tween-20 (MABT), the embryos were incubated in blocking solution (20% whole goat serum in MABT) for 2 hr; an anti-dig alkaline phosphatase-conjugated antibody (Roche) was then added to the blocking solution, and embryos were incubated at 4℃ overnight. A colour reaction was performed using NBT/BCIP substrates (Roche). To stop the reaction, embryos were fixed in 4% PFA.

## Chick sectioning and immunostaining

For wholemount immunostaining St.3–4 embryos were dissected with their yolk sac, fixed in 4% PFA for 1 hr and then subjected to standard immunostaining protocol. For cryo-sections, embryos were fixed in 4% PFA, incubated overnight in 30% sucrose in PBS at 4℃ and then embedded in a Peel-A-Way plastic mold (VWR, Radnor, PA USA) in OCT compound (Sakura, VWR). The embryos were

sectioned between 7–15 μm, using a Leica cryostat. Sections were blocked with 5% whole goat serum in 1% bovine serum albumin in PBS, prior to incubation with primary antibodies. We used the following primary antibodies: Nkx2.5 (1:300, Santa-Cruz SC-8697) was amplified using the Tyramid amplification kit (Perkin-Elmer Waltham, MA USA and Life Technologies, Carlsbad, CA USA). Isl1 (DSHB, 40.3A4, 1:5), CD45 (1:100, Prionics HIS-C7), Runx1 (1:100, Abcam ab92336), EGFP (1:100, Santa Cruz SC-101536), and a novel in-house Cdh5 (1:100, VE-cadherin). Secondary antibodies: Cy2/488, Cy3/594 and Cy5/647-conjugated anti-mouse, anti-rabbit, anti-rat IgG (Jackson ImmunoResearch laboratories, West Grove, PA USA and Thermo Fischer) were diluted 1:100.

## Chick imaging and time-lapse microscopy

Images are obtained using a Leica MZ 16FA stereomicroscope attached to a Leica digital camera (DC300F, Leica Microsystems, Wetzlar, Germany) and an assembled upright fluorescent microscope (Nikon ECLIPSE 90i) with a CCD camera. For time lapse filming a temperature controlled chamber (semi-automated) is used under the software advanced acquisition system (Image-Pro AMS version 6.0, Media Cybernetics, Bethesda, MD, USA). The system allows multi channel (bright field and GFP ) Z-stacks (for constructing a focused image) photography at multi time points for generating a time lapse movie. Cell tracking in live embryos was performed by manual analysis of 7 focal planes at each time point during *in vivo* imaging. Time lapse imaging was also performed using the DeltaVision Elite system (Applied Precision, USA), on an Olympus IX71 inverted microscope, running softWoRx 6.0 by a CoolSnap HQ2 CCD camera (Roper Scientific, USA). Still images were also obtained with the Zeiss Cell Observer Spinning Disk Confocal microscope (Oberkochen, Germany). Captured images were analyzed by ZEN, Photoshop, Image-Pro AMS and ImageJ softwares.

## *Ex ovo* electroporation

To introduce DNA vectors into the embryonic mesoderm, a method was used involving electroporation of mesodermal cells while they are still epithelial, prior to their ingression during gastrulation. Chick embryos were cultured in 'EC-culture' (*Chapman et al., 2001*). Embryos were then placed ventral side up (the vitelline membrane on the bottom) above the cathode, made of a 2 × 2 mm platinum plate located in a concavity made in the plastic platform. A solution of 1 μg/μl DNA and Fast Green (10 μg/ml) was injected between the blastoderm and the vitelline membrane, using a glass capillary. An anodal electrode, with an inter-electrode distance of 4 mm, was quickly placed on the hypoblast side of the embryo, and electroporation was performed using an ECM830 electroporator (BTX Co. Ltd, Holliston, MA USA). St. 3 embryos were electroporated with 3 pulses of 9V each, for 35 ms at intervals of 300 ms. After electroporation was complete, embryos were transferred back to the EC culture plates, and incubated to the desired stage.

## Reporter and over-expression plasmids

A novel *Isl1* cardiac enhancer was identified, using comparative genomic analysis (genomes of chick, rat, zebrafish and human were used). The enhancer was cloned into previously described pTK vectors that contain a minimal promoter from the HSV *thymidine kinase* gene upstream of GFP/RFP, and were shown to be permissive for enhancer activity in chick embryos (*Uchikawa et al., 2004*). Control vectors for broad expression within the embryo, the reporter plasmids pCAGG-GFP/RFP, which were previously described (*Nathan et al., 2008*), were injected. The mouse *Nkx2-5* cardiac enhancer was obtained from Prof. Eric Olson (*Lien et al., 1999*) and subcloned into the pTK vector for chick expression. To study the relationship between the hemangioblast and the cardiac mesoderm, we obtained a hemangioblast enhancer based on the chick *Cerberus* gene, that drives expression of GFP (*Teixeira et al., 2011*). For ectopic expression of Nkx2.5 and Tal1, chick coding sequences were cloned into pCAGG-IRES-GFP, which drives ubiquitous expression in every avian cell (*Megason and McMahon, 2002*).

## Explants and RNA analysis

Dissection of different explants from chick embryos was carried out using a tungsten needle. Explants were isolated together with all germ layers at selected embryonic stages and locations. Posterior LPM explants were cultured for 4.5 days on a collagen drop covered with 700 μl of dissection medium (10% Fetal Calf Serum, chick embryo extract 2.5% and pen/strep 1% in αMEM medium,

Biological Industries, Israel) in a four-well plate. Total RNA was extracted using Qiagene RNeasy Micro Kit (Qiagen, Germany), followed by reverse transcription using the cDNA Reverse Transcription kit (Applied Biosystems, Foster City, CA USA). The cDNA product was then amplified using different sets of primers, via semi-quantitative RT-PCR. The RNA analysis was performed using semi-quantitative RT-PCR with Green Master Mix (Promega, Madison, WI USA).

### Primers used for semi-quantitative RT-PCR

| Primer name | 5' seq | 3' seq |
| --- | --- | --- |
| Bry-qpcr | ACGCCATGTACTCCTTCCTG | TGTTGGTGAGCTTGACCTTG |
| CD31-qpcr | TGTGGAAGCAGGTGGAAAGA | CACTTCTTCTGGCAGCTCAC |
| Ets1-qpcr | TCAGTCATCCTTCGTGACCC | CCACCCAGTTTACCTCGACT |
| Flk1-qpcr | GGAGGACGCTGGTTCTGAAG | ATGTCTCGAGCCAAGCCAAA |
| Gata1-qpcr | AGACATCCACCACCACTCTG | GTTTCGGGTTTGGATTCCGT |
| Gata4-qpcr | TCCTACTCCAGCCCTTACCC | AAAAATTCTGCGATGTTGGCA |
| Isl1-qpcr | TGCAGATGGCAGCAGAACCT | TGCTCTTTCATGAGGGCGTC |
| Mesp2-qpcr | GGTCATCACCCTCCTACAGC | TCTGCATCCACAAAGTCTGG |
| Nkx2.5-qpcr | AGGCGGACAAGAAAGAACTG | CCAGTTCGTAGACTTGGGCT |
| Runx1-qpcr | TCCTACCAGTACCTGGGCTC | GTCAGAAGCACCTGAGAGGC |
| Tal1-qpcr | TCTTGCGCCTGGCTATGAAA | GAGCTTCCACAGCTGGAGTT |
| Tbx5-qpcr | GGCGAAGGAAGCTCGTAACA | ACTTTGATCCCCTCCATGCC |
| GAPDH-qpcr | AGAACATCATCCCAGCGTCCAC | ACGGCAGGTCAGGTCAACAAC |

### Analysis of single cell transcriptome data

Gene-level read counts data from single cell RNA-seq of single cells from E7-0-7.75 mouse embryos (*Scialdone et al., 2016*) were downloaded from http://gastrulation.stemcells.cam.ac.uk. Read counts were normalized for sequencing depth using size factors calculated with DESeq (*Anders and Huber, 2010*). Only samples from E7.0-E7.75 stage embryos were used. t-SNE plots were generated with highly variable genes as described (*Scialdone et al., 2016*). Single cell RNA-seq samples were categorized as expressing *Nkx2-5* if they had normalized read counts > 1/cell.

## Acknowledgements

This work was supported by grants to E.T. from the European Research Council and the Israel Science Foundation, and to RPH from the National Health and Medical Research Council of Australia (NHMRC 1074386; 573732) and the Australian Research Council Strategic Initiative in Stem Cell Science (*Stem Cells Australia* SR110001002). ET and RPH are funded by the Leducq Foundation. RPH was supported by an NHMRC Australia Fellowship (573705). PPLT is an NHMRC Senior Principal Research Fellow (1110751). We thank Karina Yaniv (Weizmann Institute of Science, Israel) for critical advice. We thank Elad Bassat (Weizmann Institute of Science, Israel) for his technical help, and Caroline and Geoffrey Burns (Harvard Stem Cell Institute, USA) for sharing unpublished data. R. P. H acknowledges the support of the Joseph Meyerhoff Visiting Professorship at the Weizmann Institute of Science.

## Additional information

### Funding

| Funder | Grant reference number | Author |
| --- | --- | --- |
| Israel Science Foundation | | Lyad Zamir |
| | | Elisha Nathan |

| | | |
|---|---|---|
| | | Oren Yifa<br>Eldad Tzahor |
| National Health and Medical Research Council | 1074386 | Reena Singh<br>Ralph Patrick<br>Patrick PL Tam<br>Richard P Harvey |
| Stem Cells Australia | SR110001002 | Reena Singh<br>Ralph Patrick<br>Richard P Harvey |
| National Health and Medical Research Council | 573732 | Reena Singh<br>Elisha Nathan<br>Oren Yifa<br>Eldad Tzahor |
| National Health and Medical Research Council | 1110751 | Reena Singh<br>Ralph Patrick<br>Patrick PL Tam |

The funders had no role in study design, data collection and interpretation, or the decision to submit the work for publication.

## Author contributions

LZ, Conceptualization, Data curation, Investigation, Methodology, Writing—original draft, Writing—review and editing; RS, Investigation, Methodology, Writing—original draft, Writing—review and editing; EN, YY-R, Methodology; RP, Data analysis; OY, Performed experiments and analyzed data; AAA, TMS, SS, J-DJH, GP, NJ, PPLT, Resources; YW, NP, Analyzing RNA-seq data; RPH, Conceptualization, Resources, Data curation, Supervision, Investigation, Methodology, Writing—original draft, Writing—review and editing; ET, Data curation, Methodology, Writing—original draft, Writing—review and editing

## Author ORCIDs

Lyad Zamir, http://orcid.org/0000-0002-3442-2952
Naihe Jing, http://orcid.org/0000-0003-1509-6378
Richard P Harvey, http://orcid.org/0000-0002-9950-9792
Eldad Tzahor, http://orcid.org/0000-0002-5212-9426

## Ethics

Animal experimentation: Mice used in this study were bred and maintained in the Victor Chang Cardiac Research Institute BioCore facility according to the Australian Code of Practice for the Care and Use of Animals for Scientific Purposes. Use of chick embryos before embryonic Day 21 does not require IACUC approval.

# Additional files

## Major datasets

The following previously published datasets were used:

| Author(s) | Year | Dataset title | Dataset URL | Database, license, and accessibility information |
|---|---|---|---|---|
| Peng G, Suo S, Chen J, Jing N, Han JJ | 2016 | Stereo-sequencing: 3D Transcriptome of the mouse embryo at gastrulation | http://www.ncbi.nlm.nih.gov/geo/query/acc.cgi?acc=GSE65924 | Publicly available at the NCBI Gene Expression Omnibus (accession no: GSE65924) |
| Scialdone A, Tanaka Y, Jawaid W, Moignard V, Wilson NK, Macaulay IC, Marioni JC, Göttgens B | 2016 | Single Cell Expression Profiling Resolves the Transcriptional Programs of Early Mesoderm Diversification | https://www.ncbi.nlm.nih.gov/geo/query/acc.cgi?acc=GSE74994 | Publicly available at the NCBI Gene Expression Omnibus (accession no: GSE74994) |

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
