## [Decision Letter]

[Editors’ note: a previous version of this study was rejected after peer review, but the authors submitted for reconsideration. The first decision letter after peer review is shown below.]

Thank you for choosing to send your work entitled "Nkx2.5 marks hemangioblast progenitors that contribute to the formation of hemogenic endothelium in the heart" for consideration at *eLife*. Your full submission has been evaluated by Janet Rossant (Senior Editor), a Reviewing Editor, and two peer reviewers, and the decision was reached after discussions between the reviewers. Based on our discussions and the individual reviews below, we regret to inform you that your work will not be considered further for publication in *eLife*.

The reviewers found the overall study to be potentially interesting and provocative, with the suggestion that both hemogenic endocardium in the heart and hemogenic endothelium in the dorsal aorta derive from Nkx2.5 expressing mesodermal cells. However, the reviewers were not convinced that the data as presented are sufficient to support this conclusion, without direct lineage tracing experiments. They also noted worrying inconsistencies between the labeling by *Nkx2.5-en* and in situ hybridization by Nkx2.5 in the LPM. At this stage, a more definitive study with different experimental approaches would be required before we could consider another version. The full reviews with suggestions for further studies are appended below.

*Reviewer #1:*

This work investigates mesodermal progenitor specification and migration during the development of the cardiovascular and blood systems. The authors use known and novel cardiovascular enhancers to activate reporter gene expression in avian embryos and follow the migration of distinct mesodermal populations over time. The most novel aspect of the study is the finding that some endocardial cells derive from a separate mesodermal origin outside the heart. More specifically, the authors propose that this extraembryonic mesoderm derived population gives rise to hemogenic endocardium, which has been recently described in the heart. This is a potentially novel finding that advances the field. The authors propose that both hemogenic endocardium in the heart and hemogenic endothelium in the dorsal aorta derive from Nkx2.5 expressing mesodermal cells ("hemangioblasts") and propose a model in which transient expression of Nkx.5 activates a hemato-vascular program, while sustained expression leads to cardiomyocyte development.

Although these findings are of potential interest and importance, it is sometimes difficult to follow the IF figures, and some data regarding the major conclusions require clarification. As an example, there is discrepancy between the enhancer activity and mRNA and protein expression that is not fully clarified. Also, the data that Nkx2.5 expressing cells form hemogenic clusters in the endocardium is not clear based on the figures. It is also not clear from the description and data how the migration of the Nkx2.5 derived cells to the heart occurs. Another potential weakness is that the paper is largely descriptive rather than mechanistic, although the authors do show that early overexpression of Nkx2.5 can induce broadly a hemato-vascular expression program. Moreover, they show that Tal1/Scl, a blood specification factor that is also induced by Nkx2.5 overexpression in this system, represses cardiomyocyte specification in a subset of mesodermal cells, although the finding that Tal1 represses cardiomyocyte development is not entirely new.

Thus, this manuscript is of potential interest, but can be fully evaluated only once these issues have been attended to.

*Reviewer #2:*

This is an interesting manuscript. The main claim of this paper is that Nkx2.5 expression marks mesodermal progenitors that contribute to the formation of hemogenic endothelium in the heart and AGM.

The data presented in this manuscript are not sufficient to support this claim. The authors demonstrate that LPM cells marked with *Nkx2.5-en* migrate to the heart and to the dorsal aorta. They also demonstrate that some cells labeled by Hb-en are also marked by *NKx2.5-en* in the LPM and that Hb-en marks hemogenic endothelium and blood in the AGM and the heart. However, there is no direct evidence that the cells co-labeled by Hb-en and *NKx2.5-en* in the LPM are the ones that contribute to blood formation in these regions. Also, the absence of correlation between the labeling by *Nkx2.5-en* and in situ hybridization by Nkx2.5 in the LPM is worrisome.

Nkx2.5 genetic lineage tracing experiments would be required to definitively prove that Nkx2.5 transient expression marks mesodermal progenitors that contribute to the formation of hemogenic endothelium in the heart and AGM. Importantly, to demonstrate that primitive streak derived hemangioblasts give rise to hemogenic endothelium in the AGM, continuous time lapse imaging should be performed.

[Editors’ note: what now follows is the decision letter after the authors submitted for further consideration.]

Thank you for submitting your article "Nkx2.5 marks angioblasts that contribute to hemogenic endothelium of the endocardium and dorsal aorta" for consideration by *eLife*. Your article has been reviewed by two peer reviewers, and the evaluation has been overseen by a Reviewing Editor and Marianne Bronner as the Senior Editor. The reviewers have opted to remain anonymous.

The reviewers have discussed the reviews with one another and the Reviewing Editor has drafted this decision to help you prepare a revised submission.

Summary:

Zamir et al. show that Nkx2.5 expressing mesodermal precursors in the lateral plate mesoderm and the yolk sac give rise to a subpopulation of endocardial cells, as well as to hemogenic endocardium and hemogenic endothelium in the dorsal aorta. The data are derived from genetic lineage tracing in mouse embryos, as well as lineage tracing via expression of a GFP-tagged mouse Nkx2.5 enhancer in chick embryos. Both approaches provide converging evidence that supports the conclusion.

Essential small, but important, revisions and minor points:

There are a number of recommendations concerning data presentation and edits that should be made before the final manuscript is accepted.

*Reviewer #1 (Minor comments):*

1) Figure 2: The images are a little difficult to interpret. Higher magnification images would support the claim that the cells in the populations are mNkx2.5-en and mlsl1-en double positive or single positive. Also, labelling regions such as cardiac crescent, area vasculosa and heart tube etc. would enhance readability.

2) Introduction, second paragraph: some references are missing and it would be more precise to write: However, while blood cell formation from "hemogenic endothelium" has been visualised directly in multiple animal models (Boisset et al., 2010; Lam et al., 2010, Bertrand 2010, Kissa 2010), and in vitro during ES cells differentiation (Eilken et al., 2009; Lancrin 2009), the existence in vivo of a bipotential hemangioblast that contributes to both blood and endothelial cells remains controversial (Hirschi, 2012).

3) I am not sure why the authors do not call their Nkx2.5 cell population a hemangioblast population as these cells in nascent mesoderm are able to form both endothelial and blood cells as stated in the abstract (although it is not shown at a clonal level). Also, the authors do not present enough data to claim that they uncovered a novel role of Nkx2.5 (Abstract and end of the Introduction), the data presented are mostly descriptive and establish mainly that Nkx2.5 is expressed in hemangioblast. The Abstract and text should be changed accordingly. I would propose for the Abstract "Taken together, we identified a hemangioblast cell lineage characterized by transient Nkx2.5 expression that contributes to hemogenic endothelium and endocardium, uncovering Nkx2.5 as a novel marker of hemangioblast lineage specification and diversification."

4) When the authors discuss in the second paragraph of the Introduction and in the fifth and eighth paragraphs of the Discussion, the migration of hemogenic endothelium precursors to different embryonic sites of hematopoiesis (Tanaka 2014), they should also mention the recent paper by Eliades 2016 (PMID:27239041). Also, it would be interesting to mention the potential role of RUNX1 in cell migration/adhesion (PMID:25082880). Finally, the meaning of the last sentence of this paragraph is not clear: Runx1 is essential at this early stage for priming the establishment of hemogenic endothelial cell fate in the aorta-gonad-mesonephros (AGM) region, where Runx1 is also activated, and for definitive hemopoiesis (Tanaka et al., 2012).

5) Please rename the video titles so that they fit the video titles in the manuscript text.

6) Figure 4. Can the authors supply single channel images to support the claim of Hb-en and mNkx2.5-en double positive cells in the endocardium?

7) Figure 5. Only a single RUNX1+/Nkx2.5 en+ cell is shown. What percentage of RUNX1+ HE cells found at this stage in the dorsal aorta are Nkx2.5en+?

8) Figure 7: Mouse Nkx2.5 lineage+ cells contribute to yolk sac vasculature. Pecam1 staining is required to confirm the endothelial nature of the Nkx2.5 lineage+ cells associated with the yolk sac vasculature.

9) Do the Nkx2.5 expressing cells detected in E7.5 murine nascent mesoderm (Figure 7—figure supplement 2) also express hematopoietic or endothelial markers?

10) Figure 8 the expression of RUNX1 and Tal1 is not obvious.

11) Discussion, first paragraph: are the authors claiming the presence of hemogenic endothelium in the liver? This is not supported by any data.

12) Subsection “Mouse Nkx2-5 lineage^+^ cells contribute to yolk sac vasculature”, last paragraph: Typo misspelling of 'lineage'

13) Subsection “Nkx2.5 is expressed in the nascent mesoderm in the primitive streak of chick embryos”, second paragraph: Typo misspelling of "immunofluorescence"

14) Subsection “Ectopic expression of Nkx2.5 induces hemangioblast gene expression”, last paragraph: Typo misspelling of 'definitive'

*Reviewer 2 (Minor comments):*

1) The authors should start the paper with the mouse data, which they then confirm using expression of the mouse enhancer in chick embryos. This would make the paper somewhat easier to read, as in the current version they switch from chick to mouse then back to chick.

2) Question regarding the mouse data: Figure 7 show that the majority of YFP cells in the DA are also Runx1+. However, this is not reflected in the quantification in Figure.7—figure supplement 1I. Please clarify.

3) Current Figure 1 starts off with an Isl1 enhancer, which is really used as a negative control. The data should be reorganized to start off with the Nkx2.5 enhancer that they use to drive GFP expression in chick embryos. Panel J reads Isl1-en I suppose this should read Nkx2.5-en. There are several J' panels. Overall, figure legends should be matched to Figures (Figure 6 and Figure 2—figure supplement 1 Figure are other examples!).

4) Do the authors claim that the mouse enhancer is sufficient to drive Nkx2.5 expression in hemogenic endothelium and endocardium? Please comment.

5) Data in Figure 3 and Figure 4 use electroporation with a GFP-tagged Cerberus hemangioblast enhancer Hb-en as a positive control. I find co-localization of Hb-en and Nkx2.5-en in Figure 3 convincing, but data in Figure 4 should track double-labeled Hb-en and Nkx2.5-en cells, or at least single labeled Nkx2.5+ cells! Likewise, explant data in Figure 3—figure supplement 1 should use both Nkx2.5-en and Hb-en.

6) Do Nkx2.5+ cells contribute some or all of the hemogenic endothelial cells in the DA and endocardium? And when and how do they get to that location? Please discuss.

---

## [Author Response]

[Editors’ note: the author responses to the first round of peer review follow.]

*Reviewer #1:*

*[…] Although these findings are of potential interest and importance, it is sometimes difficult to follow the IF figures, and some data regarding the major conclusions require clarification.*

We have improved and replaced many of the figures. In fact, Figure 4–Figure 8 and their corresponding figure supplements have been modified in the revised manuscript. Additional results were added (e.g., the contribution of the Nkx2.5 lineage in mice to both dorsal aorta (DA), liver and heart endocardium; new Figure 7 and associated supplementary figures). Labeling of the figures was also improved.

*As an example, there is discrepancy between the enhancer activity and mRNA and protein expression that is not fully clarified.*

We have expanded our temporal gene expression analysis of Nkx2.5 (RNA and protein) in both chick and mouse embryos in various mesoderm populations at different stages (Figure 7 and Figure 8, Figure 7—figure supplement 1-3, Figure 8—figure supplement 1). Importantly, whole-mount immunostaining on gastrulating St. 4 chick embryos revealed clear expression of Nkx2.5 protein in and outside the primitive streak when newly formed mesoderm cells arise (Figure 8). In addition, we have performed extensive lineage tracing experiments in mice using the Nkx2-5^IRESCre^ driver line (Figure 7, Figure 7—figure supplement 1-3), which, combined with expression analysis using Nkx2-5*^lacZ^* knockin mice, reveal the origins or extracardiac Nkx2-5 lineage^+^ cells. This has not been revealed in previous studies. These experiments performed in E7.0-10.5 embryos recapitulate most of the data obtained in the chick using the Nkx2.5 enhancer (new Figure 7, Figure 7—figure supplement 1). Although the timing of the first appearance of Nkx2-5 lineage^+^ cells in extraembryonic mesoderm differs somewhat between chick and mouse, our experiments are highly concordant. We document that Nkx2.5 expressing cells label two mesoderm populations in E7.5 mouse embryos: 1) Cardiac progenitors within the cardiac crescent and 2) angioblasts scattered throughout the LPM and yolk sac.

Importantly, at E9.5-10.5, the majority of the yolk sac vasculature as well as hemogenic endothelial cells in the DA, liver and heart could be traced to the extracardiac Nkx2.5 lineage^+^ cells in the mouse. Based on the chick and mouse data sets we conclude that the mouse extracardi Nkx2-5^+^ cells are analogous to those detected in extraembryonic/LPM cells in chick embryos and represent hemo-angiogenic precursors that later populate the endothelium of yolk sac vasculature, and hemogenic endothelium. Thus, the Nkx2.5 enhancer system in the chick serves as a valid lineage-tracing tool to label a population of mesoderm progenitors that transiently express Nkx2.5. We discuss the temporal differences between the mouse and chick data.

*Also, the data that Nkx2.5 expressing cells form hemogenic clusters in the endocardium is not clear based on the figures. It is also not clear from the description and data how the migration of the Nkx2.5 derived cells to the heart occurs.*

To address this issue we have performed additional experiments in chick embryos with the *mNkx2.5*-en. Embryos expressing the enhancer were analyzed for expression in both heart and DA. *mNkx2.5*-en^+^ cells expressing endothelial and blood markers (Cdh5, Runx1, CD45) were detected in the endocardium and DA. Additional IF panels of the *Nkx2.5 enhancer* expression along with Cdh5, Runx1, and CD45 were added see (Figure 5–Figure 6).

The migration pattern of these *mNkx2.5*-en^+^ hemogenic angioblasts could be visualized in the new supplementary movies (Video 1–Video 3). Furthermore, lineage tracing experiments in the mouse revealed the contribution of the Nkx2-5 lineage to hemogenic endothelium in the DA, and heart as shown by immunostaining for hemogenic markers (Figure 7 and Figure 7—figure supplements 1-3). Interestingly, hemogenic cells from the Nkx2.5 lineage were also detected in the liver, a bona fide organ for embryonic hematopoiesis (Figure 7).

The migration of the cells into the heart is independent of circulation as it starts around St. 5-7, prior to the formation of the primitive heart tube. Analysis of the time-lapse movies suggests that in St. 7 chick embryos there are distinct cell populations that contribute to specific parts of the heart. Cardiac progenitors within the cardiac crescent maintain their relative anterior-posterior axis in the fused heart tube [consistent with previous findings (Wei and Mikawa, 2000)].

The migration pattern of the hemo-angiogenic precursors can be seen in the revised
Figure 2—figure supplement 1 where a single cell can be traced migrating towards the inflow tract. High magnification images were added to this figure improving single cell track resolution.

*Another potential weakness is that the paper is largely descriptive rather than mechanistic, although the authors do show that early overexpression of Nkx2.5 can induce broadly a hemato-vascular expression program.*

We agree with this assessment although the primary purpose of this paper is to establish the basis for further mechanistic studies from careful lineage tracing data. Having said that, the gain of function data presented in Figure 9 as well as in Figure 9—figure supplement 1 and Figure 9—figure supplement 2, are novel and striking, and establish that transient expression of Nkx2.5 in the primitive streak in chick is likely to be functionally important for hemoangiogenesis. We show that Nkx2.5 overexpression induces a set of endothelial/blood markers including *Tal1, Flk1, Ets1 and Lmo2*. Hence Nkx2.5 can induce hemato- vascular gene expression in-vivo.

*Moreover, they show that Tal1/Scl, a blood specification factor that is also induced by Nkx2.5 overexpression in this system, represses cardiomyocyte specification in a subset of mesodermal cells, although the finding that Tal1 represses cardiomyocyte development is not entirely new.*

We agree that these results support the loss of function studies of Tal1/Scl in mice, in which there was ectopic cardiogenesis (Schoenebeck et al., 2007; Van Handel et al., 2012). However, to our knowledge, the effect of Tal1 overexpression in vivoon cardiac development has not been described before. The perturbation of the cardiovascular system by Tal1 overexpression was manifested by a set of defects in heart tube formation leading to edema. The effect of Tal1/Scl on cardiogenesis is still unclear as a very recent Nature paper argues against the repressive role of Scl1/Tal1 on cardiogenesis (Scialdone et al., 2016). However, as noted, our experiment is complementary to the genetic loss of function data in mice. We included these data as supplementary information in order to validate the specificity of, and contrast the novel findings obtained in, the Nkx2.5 overexpression experiment.

*Reviewer #2:*

*This is an interesting manuscript. The main claim of this paper is that Nkx2.5 expression marks mesodermal progenitors that contribute to the formation of hemogenic endothelium in the heart and AGM.*

*The data presented in this manuscript are not sufficient to support this claim. The authors demonstrate that LPM cells marked with Nkx2.5-en migrate to the heart and to the dorsal aorta. They also demonstrate that some cells labeled by Hb-en are also marked by NKx2.5-en in the LPM and that Hb-en marks hemogenic endothelium and blood in the AGM and the heart. However, there is no direct evidence that the cells co-labeled by Hb-en and NKx2.5-en in the LPM are the ones that contribute to blood formation in these regions. Also, the absence of correlation between the labeling by Nkx2.5-en and in situ hybridization by Nkx2.5 in the LPM is worrisome.*

*Nkx2.5 genetic lineage tracing experiments would be required to definitively prove that Nkx2.5 transient expression marks mesodermal progenitors that contribute to the formation of hemogenic endothelium in the heart and AGM. Importantly, to demonstrate that primitive streak derived hemangioblasts give rise to hemogenic endothelium in the AGM, continuous time lapse imaging should be performed.*

We thank the reviewer for his assessment of the work.

As indicated above and in response to this reviewer’s comments we now provide additional experiments in support of our work in chick embryos by adding an extensive set of lineage tracing experiments in the mouse done in collaboration with Richard Harvey’s lab demonstrating the contribution of extra-cardiac Nkx2.5 lineage^+^ cells to the yolk-sac vasculature as well as to the hemogenic endothelium in the dorsal aorta and the liver (Figure 7) as well as to the endocardium (Figure 7—figure supplement 1). Sensitive expression analysis using Nkx2-5^LacZ^ knockin mice, and embryo topographical RNAseq data (in collaboration with Patrick Tam and colleagues) were also performed. Regarding the lack of correlation between the enhancer and protein expression, we now show Nkx2.5 immunostaining demonstrating expression in the nascent mesoderm in and outside the primitive streak of St. 4 chick embryos (Figure 8). We hope that the lineage tracing and expression data in the mouse coupled with the new protein expression data showing the presence of Nkx2.5 in the nascent mesoderm of gastrulating chick embryos, addresses the major concerns of this reviewer.

For technical reasons, we could not show a single cell labeled by both enhancers in the LPM that goes on and form blood cells. However, we do show that mouse extra-cardiac Nkx2-5 lineage^+^ cells form circulating blood, confirming the data of Nakano et al. who also showed that such lineage positive cell form single and multilineage erythroid and myeloid colonies in vitro, and erythrocytes expressing both primitive and definitive hemoglobins in vivo. In this study, the origin of these blood cells was attributed to the hemogenic endocardium; however our new lineage tracing data strongly suggest that they arise from hemogenic endothelium of the yolk sac, AGM region of the dorsal aorta, or liver. No such lineage^+^ cells were found in the adult in the Nakano study.

For the benefit of the reviewers, in a preliminary screen we also assessed whether Nkx2-5 lineage^+^ cells could be found in adult bone marrow and spleen. Results indicate that 1 in 20 mice screened showed significant contribution to both organs but zero contribution in the other 19 mice. Positive controls (germline deletion of the Cre reporter) showed the expected high contribution to bone marrow and spleen. We will not present this data in the revised manuscript because only a single positive mouse was found and because individual blood lineages and serial transplantation studies were not assessed. Nonetheless the results serve to suggest that extra-cardiac Nkx2-5 lineage^+^ cells can contribute to adult blood lineages, albeit rarely, so not in a physiologically significant way. We believe that the Nkx2-5 lineage+ cells contribute to hemopoiesis only in the early stages of development.

We would like to add that our work reveals, from a cardiac development perspective, a novel mesodermal progenitor population, which is not part of the FHF or SHF. We provide compelling evidence that this population of cardiovascular progenitors shares numerous characteristics with hemogenic angioblasts (including gene expression and enhancer expression patterns, migration pattern, differentiation in culture, and in vivo cluster localization in the DA and the heart). We feel that these findings in addition to the lineage tracing and expression data in mouse embryos are novel and important, and we believe are of significant importance to the cardiac, vascular and hemopoietic fields. Continuous time lapse videos for more than 24-30 hours is technically impossible given the current tools due to the development and turning of the chick embryo. Our current technique is limited to tracing cells up to St. 12. At later stages the chick embryo becomes opaque and the position of the heart changes dramatically.

To further address the question related to the actual dynamic expression of Nkx2.5 and its enhancer we have taken several additional approaches, as noted in our response to reviewer 1.

[Editors' note: the author responses to the re-review follow.]

*Essential small, but important, revisions and minor points:*

*There are a number of recommendations concerning data presentation and edits that should be made before the final manuscript is accepted.*

*Reviewer #1 (Minor comments):*

*1) Figure 2: The images are a little difficult to interpret. Higher magnification images would support the claim that the cells in the populations are mNkx2.5-en and mlsl1-en double positive or single positive. Also, labelling regions such as cardiac crescent, area vasculosa and heart tube etc. would enhance readability.*

The figure has been revised according to the reviewers’ suggestions. Higher magnification panels were added to the figure and key anatomical structures are now marked.

*2) Introduction, second paragraph: some references are missing and it would be more precise to write: However, while blood cell formation from "hemogenic endothelium" has been visualised directly in multiple animal models (Boisset et al., 2010; Lam et al., 2010, Bertrand 2010, Kissa 2010), and* in vitro *during ES cells differentiation (Eilken et al., 2009; Lancrin 2009), the existence* in vivo *of a bipotential hemangioblast that contributes to both blood and endothelial cells remains controversial (Hirschi, 2012).*

We modified the sentence according to the suggestion. These relevant papers are now cited in the text. We thank the reviewer for this.

*3) I am not sure why the authors do not call their Nkx2.5 cell population a hemangioblast population as these cells in nascent mesoderm are able to form both endothelial and blood cells as stated in the abstract (although it is not shown at a clonal level). Also, the authors do not present enough data to claim that they uncovered a novel role of Nkx2.5 (Abstract and end of the Introduction), the data presented are mostly descriptive and establish mainly that Nkx2.5 is expressed in hemangioblast. The Abstract and text should be changed accordingly. I would propose for the Abstract "Taken together, we identified a hemangioblast cell lineage characterized by transient Nkx2.5 expression that contributes to hemogenic endothelium and endocardium, uncovering Nkx2.5 as a novel marker of hemangioblast lineage specification and diversification."*

Basically, we agree with the reviewer that our study mainly deals with the expression of Nkx2.5 in hemangioblasts. More mechanistic data such as the knockout of Nkx2.5 in endothelial/blood lineages would be important for future studies. The severe defects in early yolk sac vasculature observed in the Nkx2.5 null embryos coupled to our gain of function experiments in the chick, as well as other recent studies support the idea that Nkx2.5 is more than “just a marker” for hemangioblasts. We have modified the Abstract along the reviewer suggestion: “we identified a hemogenic angioblast cell lineage characterized by transient Nkx2.5 expression that contributes to hemogenic endothelium and endocardium, suggesting a novel role for Nkx2.5 in hemoangiogenic lineage specification and diversification”. We have retained reference to Nkx2.5 expression in the “hemoangiogenic lineage”, avoiding specific reference to hemangioblasts, as this historical term infers a bipotential progenitor, the presence of which in vivo remains controversial, as discussed (Nishikawa, 2012).

*4) When the authors discuss in the second paragraph of the Introduction and in the fifth and eighth paragraphs of the Discussion, the migration of hemogenic endothelium precursors to different embryonic sites of hematopoiesis (Tanaka 2014), they should also mention the recent paper by Eliades 2016 (PMID:27239041). Also, it would be interesting to mention the potential role of RUNX1 in cell migration/adhesion (PMID:25082880). Finally, the meaning of the last sentence of this paragraph is not clear: Runx1 is essential at this early stage for priming the establishment of hemogenic endothelial cell fate in the aorta-gonad-mesonephros (AGM) region, where Runx1 is also activated, and for definitive hemopoiesis (Tanaka et al., 2012).*

We now discuss these references and simplified the above sentence.

*5) Please rename the video titles so that they fit the video titles in the manuscript text.*

The video titles now match the titles in the manuscript.

*6) Figure 4. Can the authors supply single channel images to support the claim of Hb-en and mNkx2.5-en double positive cells in the endocardium?*

Figure 4 was updated with the requested images. Figure 4 now includes panels 4H’-H’’’ with high magnifications of single channels for each enhancer.

*7) Figure 5. Only a single RUNX1+/Nkx2.5 en+ cell is shown. What percentage of RUNX1+ HE cells found at this stage in the dorsal aorta are Nkx2.5en+?*

Due to the transient and mosaic nature of the electroporation method in the chick it is more difficult to detect *mNkx2.5*-en^+^ (GFP^+^) cells in the dorsal aorta in comparison to genetic labeling in mice. This may lead to an underestimation of GFP^+^ cells as the embryo develops. Still, we counted GFP^+^ cells in the dorsal aorta and found that ~15% of Runx1^+^ cells in this site are GFP^+^ (Runx1^+^/*mNkx2.5*-en^+^). Strikingly, this number is in line with the mouse lineage tracing data of *Nkx2.5*-derived angioblasts expressing Runx1 (~15-16%, Figure 9—figure supplement 1). We discuss this point in the text of Figure 9 (subsection “Mouse Nkx2-5 lineage^+^ cells contribute to the hemogenic endothelium of the dorsal aorta”.

*8) Figure 7: Mouse Nkx2.5 lineage+ cells contribute to yolk sac vasculature. Pecam1 staining is required to confirm the endothelial nature of the Nkx2.5 lineage+ cells associated with the yolk sac vasculature.*

We were unable to detect Pecam1 staining in yolk sac vessels at E9.0 and suspect that these immature vessels do not express high levels of this mature endothelial cell marker. In order to confirm that Nkx2-5^+^ lineage traced cells contribute to the endothelium of the yolk sac vasculature, co-immunostaining was performed at E9.0 with the endothelial markers Vegfr2/Flk1 and VE-cadherin, detecting the Nkx2-5 lineage^+^ cells using two different Cre reporter lines *R26R^YFP^*and *Z/EG*. The majority of Nkx2-5-lineage traced YFP/GFP-positive cells lining the yolk sac vessels co-stained with Vegfr2 and VE-cadherin, suggesting contribution to the endothelium of these vessels (see new textin subsection “Mouse Nkx2-5 lineage^+^ cells contribute to yolk sac vasculature”, first paragraph and Figure 9—figure supplement 2).

*9) Do the Nkx2.5 expressing cells detected in E7.5 murine nascent mesoderm (*Figure 7—figure supplement 2*) also express hematopoietic or endothelial markers?*

To address this question, we have exploited the extensive single cell transciptome data of Berthold Gottgens and colleagues (Scialdone et al., 2016). In this work, they profiled the transcriptome of single cells from E6.5 epiblast, as well as Flk1^+^ mesodermal cells from E7.0-E7.75 embryos. Epiblast cells, extraembryonic ectoderm and some 7 different mesodermal cell types were identified using the cell surface markers Flk1 and CD41, as well as the transcriptome data. We interrogated the single cell data in E7.0-E7.75 embryos for *Nkx2-5* expression. While expression was not detected in epiblast or extra-embryonic ectodermal cells, expression was evident in ~7% of nascent mesodermal cells and ~30% of posterior mesoderm and pharyngeal mesoderm. Notably, ~30% of vascular endothelial cells and allantois (one extraembryonic site of hemopoiesis), and ~10% of early blood progenitor were also positive. Thus, as suggested by immunofluorescence, *Nkx2-5* is expressed in endothelial cells and blood progenitors in early murine embryos. This data is discussed in the last paragraph of the subsection “Mouse Nkx2-5 lineage^+^ cells contribute to yolk sac vasculature” and shown in Figure 9—figure supplement 3.

*10) Figure 8 the expression of RUNX1 and Tal1 is not obvious.*

In the text we now note that the expression of these genes is very low.

*11) Discussion, first paragraph: are the authors claiming the presence of hemogenic endothelium in the liver? This is not supported by any data.*

Our data in mice reveal that a proportion of cells expressing HSC markers Runx1^+^ and CD41^+^ in the fetal liver are derived from the Nkx2.5^+^ lineage (Figure 9). Significant numbers of co-labeled cells were observed in the liver at E10.5. Characterization of these liver progenitors would require further studies. The text was revised to make this point clear.

*12) Subsection “Mouse Nkx2-5 lineage^+^ cells contribute to yolk sac vasculature”, last paragraph: Typo misspelling of 'lineage'*

Fixed.

*13) Subsection “Nkx2.5 is expressed in the nascent mesoderm in the primitive streak of chick embryos”, second paragraph: Typo misspelling of "immunofluorescence"*

Fixed.

*14) Subsection “Ectopic expression of Nkx2.5 induces hemangioblast gene expression”, last paragraph: Typo misspelling of 'definitive'*

Fixed and many thanks for these corrections.

*Reviewer 2 (Minor comments):*

*1) The authors should start the paper with the mouse data, which they then confirm using expression of the mouse enhancer in chick embryos. This would make the paper somewhat easier to read, as in the current version they switch from chick to mouse then back to chick.*

We thank the reviewer for this suggestion but we feel that the story flows nicely as is starting with the chick data that identified a population of hemangioblasts labeled by the *Nkx2.5* enhancer. We then go on to confirm the data in mice using *Nkx2-5*-LacZ expression and lineage tracing methods. We rearranged the figures such that all the chick data now precede the mouse data to avoid the back and forth situation between the two models.

*2) Question regarding the mouse data: Figure 7 show that the majority of YFP cells in the DA are also Runx1+. However, this is not reflected in the quantification in Figure 7—figure supplement 1. Please clarify.*

Please note that formerly Figure 7 and Figure 7—figure supplement 1 are now Figure 9 and Figure 9—figure supplement 1, respectfully.

We understood that reviewer’s comments to mean that in Figure 9’’’, few Runx1- YFP^+^ cells are present in the sections shown, whereas in Figure 9—figure supplement 1, some 23% of endothelial cells were scored as Runx1^-^ YFP^+^. It is true that few such endothelial cells were present in the sections selected for Figure 9’’’, and we have now added a new set of panels to Figure 9—figure supplement 1’’’ to show a region in which there are more Runx1^-^ YFP^+^ cells (highlighted by green arrows), and referenced this in the first paragraph of the subsection “Mouse Nkx2-5 lineage^+^ cells contribute to the hemogenic endothelium of the dorsal aorta”.

Please note that the quantification data shown in Figure 9—figure supplement 1 are robust – fifteen 10 μm sections were analyzed from three different *Nkx2-5^irescre^; R26R^YFP^* embryos. A total of 859 cells were scored.

*3) Current Figure 1 starts off with an Isl1 enhancer, which is really used as a negative control. The data should be reorganized to start off with the Nkx2.5 enhancer that they use to drive GFP expression in chick embryos. Panel J reads Isl1-en I suppose this should read Nkx2.5-en. There are several J' panels. Overall, figure legends should be matched to Figures (Figure 6 and Figure 2—figure supplement 1 are other examples!).*

We thank the reviewer for his suggestion and we modified Figure 1 accordingly. The change is also reflected in the text. Corrections to figure legends and within the figures were also made. Thanks again!

*4) Do the authors claim that the mouse enhancer is sufficient to drive Nkx2.5 expression in hemogenic endothelium and endocardium? Please comment.*

We argue that the mouse enhancer that we used in the chick embryos is sufficient to drive the reporter expression in a special population of hemogenic angioblasts that could be found in the hemogenic endothelium and endocardium. We could show that this occurs due to a transient expression of Nkx2.5 at RNA and protein expression in the nascent mesoderm during gastrulation. The mouse data support this conclusion, although the onset of expression in mouse occurs later. We could not show Nkx2.5 protein expression in hemogenic endothelium as levels are presumably downregulated by the time that angioblast cells adopt hemogenic characteristics, likely by a mutual inhibition of hemogenic transcription factors (e.g., Scl1/Tal1).

*5) Data in Figure 3 and Figure 4 use electroporation with a GFP-tagged Cerberus hemangioblast enhancer Hb-en as a positive control. I find co-localization of Hb-en and Nkx2.5-en in Figure 3 convincing, but data in Figure 4 should track double-labeled Hb-en and Nkx2.5-en cells, or at least single labeled Nkx2.5+ cells! Likewise, explant data in Figure 3—figure supplement 1 should use both Nkx2.5-en and Hb-en.*

Following the reviewer’s request we performed additional experiments to address the above issues. We have added new data of a time-lapse analysis performed on *mNxk2.5* and *Hb* enhancers expressing embryo (Figure 4—figure supplement 1, Video 4). In this experiment we could track double positive cells (*mNkx2.5*-en^+^/*Hb*-en^+^) that migrate into heart tube from St.7 to St. 11. In addition we detected numerous double positive cells that contribute to the extraembryonic vasculature in line with the mouse lineage tracing data.

In addition, we performed a new LPM explant assay of embryos electroporated with the *mNkx2.5*-en plasmid. We now show that a subpopulation of *mNkx2.5*-en^+^ cells in the LPM has a hemogenic potential highlighted by their Runx1 expression (Figure 3—figure supplement 1).

*6) Do Nkx2.5+ cells contribute some or all of the hemogenic endothelial cells in the DA and endocardium? And when and how do they get to that location? Please discuss.*

As stated above and based on both model systems, about 15% of the Nkx2.5 expressing cells in the LPM/extraembryonic mesoderm adopt a hemogenic angioblast characteristic, as judged by marker expression. Our methodologies could not adequately address whether all hemogenic endothelium in the DA and endocardium originate from Nkx2.5 expressing (or lineage derived) cells. Conceivably, hemogenic endothelial cells have multiple origins but this would require further studies. We added a sentence regarding this aspect in the Discussion.